# Process-oriented models of autumn leaf phenology: ways to sound calibration and implications of uncertain projections

Michael Meier[1,2], Christof Bigler[1]
1) Forest Ecology, Department of Environmental Systems Science, ETH Zurich, Zurich, Switzerland
2) CEFE, Univ. Montpellier, CNRS, EPHE, IRD, Montpellier, France

*Correspondence to*: Michael Meier (michael.meier@cefe.cnrs.fr)

**Abstract.** Autumn leaf phenology marks the end of the growing season, during which trees assimilate atmospheric $CO_2$. The length of the growing season is affected by climate change because autumn phenology responds to climatic conditions. Thus, the timing of autumn phenology is often modelled to assess possible climate change effects on future $CO_2$ mitigating capacities and species compositions of forests. Projected trends have been mainly discussed with regards to model performance and climate change scenarios. However, there has been no systematic and thorough evaluation of how performance and projections are affected by the calibration approach. Here, we analyzed >2.3 million performances and 39 million projections across 21 process-oriented models of autumn leaf phenology, 5 optimization algorithms, ≥7 sampling procedures, and 26 climate model chains from two representative concentration pathways. Calibration and validation were based on >45 000 observations for beech, oak, and larch from 500 Central European sites each. Phenology models had the largest influence on model performance. The best performing models were (1) driven by daily temperature, day length, and partly by seasonal temperature or spring leaf phenology and (2) calibrated with the Generalized Simulated Annealing algorithm (3) based on systematically balanced or stratified samples. Autumn phenology was projected to shift between −13 and +20 days by 2080–2099 compared to 1980–1999. Climate scenarios and sites explained more than 80% of the variance in these shifts and thus had an influence eight to 22 times greater than the phenology models. Warmer climate scenarios and better performing models predominantly projected larger backward shifts than cooler scenarios and poorer models. Our results justify inferences from comparisons of process-oriented phenology models to phenology-driving processes and we advocate species-specific models for such analyses and subsequent projections. For sound calibration, we recommend a combination of cross-validations and independent tests, using randomly selected sites from stratified bins based on mean annual temperature and average autumn phenology, respectively. Poor performance and little influence of phenology models on autumn phenology projections suggest that current models are overlooking relevant drivers. While the uncertain projections indicate an extension of the growing season, further studies are needed to develop models that adequately consider the relevant processes for autumn phenology.

**Key words.** Climate change, deciduous trees, leaf senescence, optimization algorithms, sampling procedures, site-specific calibration, species-specific calibration, tree phenology.

**Summary**

This study analyzed the impact of process-oriented models, optimization algorithms, calibration samples and climate scenarios on the simulated timing of autumn leaf phenology (Figure 2). The accuracy of the simulated timing was assessed by the root mean square error (RMSE) between observed and simulated timing of autumn phenology. The future timing was expressed as projected shift between 1980–1999 and 2080–2099 ($\Delta_{100}$). While the RMSE was related to the models, optimization algorithms, and calibration samples through linear mixed-effects models (LMM), $\Delta_{100}$ was related to the climate change scenarios, models, optimization algorithms, and calibration samples. The analyzed >2.3 million RMSE and 39 million $\Delta_{100}$ were derived from site- and species-specific calibrations (i.e. one set of parameters per site and species vs. one set of parameters per species, respectively). The calibrations were based on 17 211, 16 954, and 11 602 observed site-years for common beech (*Fagus sylvatica* L.), pedunculate oak (*Quercus robur* L.), and European larch (*Larix decidua* MILL.), respectively, which were recorded at 500 Central European sites per species.

*Process-oriented models are a useful tool to study leaf senescence*

The assessed phenology models differed in their functions and drivers, which had the largest influence on the accuracy of the simulated autumn phenology (i.e. model performance). In all 21 models, autumn phenology occurs when a threshold related to an accumulated daily senescence rate is reached. While the threshold is either a constant or depends linearly on one or two seasonal drivers, the rate depends on daily temperature and, in all but one model, on day length. Depending on the model, the rate is (1) a monotonically increasing response to cooler days and (i) amplified or (ii) weakened by shorter days, or (2) a sigmoidal response to both cooler and shorter days. In the three most accurate models, the threshold was either a constant or derived from the timing of spring leaf phenology (site-specific calibration) or the average temperature of the growing season (species-specific calibration). Further, the daily rate of all but one of these models was based on monotonically increasing curves, which were both amplified or weakened by shorter days. Overall, the relatively large influence of the models on the performance justifies inferences from comparisons of process-oriented models to the leaf senescence process.

*Chosen optimization algorithms must be carefully tuned*

The choice of the optimization algorithm and corresponding control settings had the second largest influence on model performance. The models were calibrated with five algorithms (i.e., Efficient Global Optimization based on kriging with/without Trust-Region formation, Generalized Simulated Annealing, Particle Swarm Optimization, and Covariance Matrix Adaptation with Evolutionary Strategies), each executed with few and many iterations. In general, Generalized Simulated Annealing found the parameters that led to the best performing models. Depending on the algorithm, model performance increased with more iterations for calibration. The positive and negative effects of more iterations on subsequent model performance relativize the comparison of algorithms in this study and exemplify the importance of carefully tuning the chosen algorithm to the studied search space.

*Stratified samples result in most accurate calibrations*

Model performance was relatively little influenced by the choice of the calibration sample in both the site- and species-specific calibration. The models were calibrated and validated with site-specific five-fold cross-validation as well as with species-specific calibration samples that contained 75% randomly assigned observations from between 2 and 500 sites and corresponding validation samples that contained the remaining observations of these sites or of all sites of the population. For the site-specific cross-validation, observations were selected in a random or systematic procedure. The random procedure assigned the observations randomly. For the systematic procedure, observations were first ordered based on year, mean annual temperature (MAT), or autumn phenology date (AP). Thus, every $5^{th}$ observation [i.e., $1+i$, $6+i,\ldots$ with $i \in (0,1,\ldots,4)$; systematically balanced] or each fifth of the $n$ observations [i.e.; $1+i,2+i,\ldots,n/5+i$ with $i \in (0,1/5\times n,\ldots,4/5\times n)$; systematically continuous] was assigned to one of the cross-validation samples. For the species-specific calibration, sites were selected in a random, systematic, or stratified procedure. The random procedure randomly assigned 2, 5, 10, 20, 50, 100, or 200 sites from the entire or half of the population according to the average MAT or average AP. For the systematic procedure, sites were first ordered based on average MAT or average AP. Thus, every $j^{th}$ site was assigned to a particular calibration sample with the greatest possible difference in MAT or AP between the 2, 5, 10, 20, 50, 100, or 200 sites. For the stratified procedure, the ordered sites were separated into 12 or 17 equal-sized bins based on MAT or AP, respectively (i.e. the smallest possible size that led to at least one site per bin). Thus, one site per bin was randomly selected and assigned to a particular calibration sample. The effects of these procedures on model performance were analyzed together with the effect of sample size. The results show that at least nine observations per free model parameter (i.e. the parameters that are fitted during calibration) should be used, which advocates the pooling of sites and thus species-specific models. These models likely perform best when (1) sites are selected in a stratified procedure based on MAT for (2) a cross-validation with systematically balanced observations based on site and year, and their performance (3) should be tested with new sites selected in a stratified procedure based on AP.

*Projections of autumn leaf phenology are highly uncertain*

Projections of autumn leaf phenology to the years 2080–2099 were mostly influenced by the climate change scenarios, whereas the influence of the phenology models was relatively small. The analyzed projections were based on 16 and 10 climate model chains (CMC) that assume moderate vs. extreme future warming, following the Representative Concentration Pathways (RCP) 4.5 and 8.5, respectively. Under more extreme warming, the projected autumn leaf phenology occurred 8–9 days later than under moderate warming, namely shifting by −4 to + 20 days (RCP 8.5) vs. −13 to +12 days (RCP 4.5). While autumn phenology was projected to generally occur later according to the better performing models, the projections were over six times more influenced by the climate scenarios than by the phenology models. This small influence of models that differ in their functions and drivers indicates that the modelled relationship between warmer days and slowed senescence rates suppresses the effects of the other drivers considered by the models. However, because some of these drivers are known to considerably influence autumn phenology, the lack of corresponding differences between projections of current phenology models underscores their uncertainty rather than the reliability of these models.

# 1    Introduction

Leaf phenology of deciduous trees describes the recurrent annual cycle of leaf development from bud set to leaf fall (Lieth, 1974). In temperate and boreal regions, spring and autumn leaf phenology divide this cycle into a photosynthetically active and inactive period, hence forth referred to as the growing and dormant season (Lang et al., 1987; Maurya and Bhalerao, 2017). The response of leaf phenology to climate change affects the length of the growing season and thus the amount of atmospheric $CO_2$ taken up by trees (e.g., Richardson et al., 2013; Keenan et al., 2014; Xie et al., 2021), as well as species distribution and species composition (e.g., Chuine and Beaubien, 2001; Chuine, 2010; Keenan, 2015). While several studies found spring phenology to advance due to climate warming (e.g., Fu et al., 2014a; Meier et al., 2021), findings regarding autumn phenology are more ambiguous (Piao et al., 2019; Menzel et al., 2020) but tend to indicate a backward shift (e.g., Bigler and Vitasse, 2021; Meier et al., 2021).

Various models have been used to study leaf phenology and provide projections, which may be grouped in correlative and process-oriented models (Chuine et al., 2013). Both types of models have served to explore possible underlying processes (e.g., Xie et al., 2015; Lang et al., 2019). The former models have often been used to analyze the effects of past climate change on leaf phenology (e.g., Asse et al., 2018; Meier et al., 2021), while the latter models have usually been applied to study the effects of projected climate change (e.g., Morin et al., 2009; Zani et al., 2020). Popular representatives of the correlative models applied in studies on leaf phenology are based on linear mixed-effects models and generalized additive models (e.g., Xie et al., 2018; Menzel et al., 2020; Meier et al., 2021; Vitasse et al., 2021), while the many different process-oriented phenology models all go back to the growing-degree day model (Chuine et al., 2013; Chuine and Régnière, 2017; Fu et al., 2020) of Réaumur (1735).

Different process-oriented models rely on different assumptions regarding the driving processes of leaf phenology (e.g., Meier et al., 2018; Chuine et al., 1999), but their functionality is identical. Process-oriented leaf phenology models typically consist of one or more phases during which daily rates of relevant driver variables are accumulated until a corresponding threshold is reached (Chuine et al., 2013; Chuine and Régnière, 2017). The rate usually depends on daily meteorological drivers and sometimes on day length (Chuine et al., 2013; Fu et al., 2020), while the threshold either is a constant (Chuine et al., 2013), depends on latitude (Liang and Wu, 2021) or on seasonal drivers (e.g. the timing of spring phenology with respect to autumn phenology; Keenan and Richardson, 2015).

Models of spring phenology regularly outcompete models of autumn phenology by several days when assessed by the root mean square error between observed and modelled dates (4–9 vs. 6–13 days, respectively; Basler, 2016; Liu et al., 2020). These errors have been interpreted in different ways and have multiple sources. Basler (2016) compared over 20 different models and model combinations for spring leaf phenology of trees. He concluded that the models underestimated the inter-annual variability of observed dates of spring leaf phenology and were not transferable between sites. Liu et al. (2020) compared six models of autumn leaf phenology of trees and concluded that the inter-annual variability was well represented by the models, while their representation of the inter-site variability was relatively poor.

Well-calibrated models of autumn leaf phenology are a prerequisite for sound conclusions about phenology-driving processes and for reducing uncertainties in phenological projections under distant climatic conditions. Studies of leaf phenology models generally show that certain models lead to better results and thus conclude that these models consider the relevant phenology-driving processes more accurately or add an important piece to the puzzle (Delpierre et al., 2009; Keenan and Richardson, 2015; Lang et al., 2019; Liu et al., 2019; Zani et al., 2020). Such conclusions can be assumed to be stronger if they are based on sound calibration and validation. However, so far, different calibration and validation methods have been applied (e.g., species- or site-specific calibration; Liu et al., 2019; Zani et al., 2020), which makes the comparison of study results difficult. Moreover, the uncertainty in leaf phenology projections is related to both climate projections and phenology models. While the uncertainty associated with climate projections has been extensively researched (e.g., Palmer et al., 2005; Foley, 2010; Braconnot et al., 2012), so far the uncertainty associated with process-oriented phenology models has only been described in a few notable studies: Basler (2016) compared spring phenology models calibrated per species and per site as well as calibrated per species with pooled sites, Liu et al. (2020) compared autumn phenology models with a focus on inter-site and inter-annual variability, and Liu et al. (2021) focused on sample size and observer bias in observations of spring and autumn phenology. Therefore, this uncertainty and its drivers are arguably largely unknown and thus poorly understood, which may be part of the reason for debates such as the one surrounding the Zani et al. (2020) study (Norby, 2021; Zani et al., 2021; Lu and Keenan, 2022).

When considering phenology data from different sites, one must, in principle, decide between two calibration modes, namely a calibration per site and species or a calibration over various sites with pooled data per species. While the former calibration leads to a set of parameters per species and site, the latter leads to one set of parameters per species. On the one hand, site-specific models may respond to local adaptation (Chuine et al., 2000) without explicitly considering the underlying processes as well as to relevant but unconsidered drivers. For example, a model based solely on temperature may provide accurately modelled data due to site-specific thresholds, even if the phenological observations at some sites are driven by additional variables such as soil water balance. On the other hand, species-specific models may consider local adaptation via parameters such as day length (Delpierre et al., 2009) and may be better suited for projections to other sites and changed climatic conditions, as they apply to the whole species and follow a space-for-time approach (but see Jochner et al., 2013).

Independent of the calibration mode, various optimization algorithms have been used for the calibration of the model parameters. The resulting parameters are often intercorrelated (e.g. the base temperature for the growing degree days function and the corresponding threshold value to reach) and the parameter space may have various local optima (Chuine and Régnière, 2017). To calibrate phenology models, different optimization algorithms have been applied to locate the global optimum, such as Simulated Annealing, Particle Swarm Optimization, or Bayesian optimization methods (e.g., Chuine et al., 1998; Liu et al., 2020; Zhao et al., 2021). Simulated Annealing and its derivatives seem to be used the most in the calibration of process-oriented models of tree leaf phenology (e.g., Chuine et al., 1998; Basler, 2016; Liu et al., 2019; Zani et al., 2020). However, a systematic comparison of these different optimization algorithms regarding their influence on model performance and projections has been missing so far.

Previous studies on process-oriented phenology models have generally provided little information on the sampling procedure used to assign observations to the calibration and validation samples. Observations and sites may be sampled according to different procedures, such as random, stratified, or systematic sampling (Taherdoost, 2016). In contrast to random sampling, systematic and stratified sampling require a basis to which the systematic or stratification refers to. For example, when assigning observations based on year, observations from every $i^{th}$ year or one randomly selected observation from each of the $i$ bins with equal time spans may be selected in systematic or stratified sampling, respectively. Studies on phenology models have usually considered all sites of the underlying dataset and declared the size of calibration and validation samples or the number of groups ($k$) in a $k$-fold cross-validation (e.g., Delpierre et al., 2009; Basler, 2016; Meier et al., 2018). However, the applied sampling procedure has not always been specified, but there are notable exceptions, such as Liu et al. (2019) for random sampling, Chuine et al. (1998) for systematic sampling, and Lang et al. (2019) for leave-one-out cross-validation. Moreover, the effects of the sampling procedure on the performance and projections of phenology models have not been studied yet.

Sample size in terms of the number of observations per site and the number of sites may influence the quality of phenology models as well. Studies on phenology models have usually selected sites with at least 10 or 20 observations per site, independent of the calibration mode (e.g., Delpierre et al., 2009; Keenan and Richardson, 2015; Lang et al., 2019). In studies with species-specific models, a wide range in the number of sites have been considered, namely 8 to >800 sites (e.g., Liu et al., 2019; Liu et al., 2020). In site-specific calibration, the number of sites may be neglected as the site-specific models cannot be applied to other sites. However, the number of observations is crucial, as small samples may lead to overfitted models due to the bias-variance trade-off (James et al., 2017, Ch. 2.2.2), i.e. the trade-off between minimizing the prediction error in the validation sample versus the variance of the estimated parameters in the calibrated models. To our knowledge, no study to date has examined possible overfitting in phenology models. In addition, in species-specific calibration, the number of sites could influence the degree to which the population is represented by the species-specific models. While such reasoning appears intuitively right, we are unaware of any study that has systematically researched the correlation between the number of sites and the degree of representativeness.

Phenology models are typically calibrated, their performance is estimated, and some studies project leaf phenology under distant climatic conditions. The performance of phenology models has often been estimated with the root mean square error that is calculated from modelled and observed data (e.g., Delpierre et al., 2009; Lang et al., 2019) and has been generally used for model comparison (e.g., Basler, 2016; Liu et al., 2020) and model selection (e.g., Liu et al., 2019; Zani et al., 2020). When phenology has been subsequently projected under distant climatic conditions, projections may have been compared between models (Zani et al., 2020), but no correlation with model performance has been established yet.

With this study, we take a first step towards closing the gap of unknown uncertainties associated with process-oriented models of autumn tree leaf phenology, which has been left open by current research so far. We focused on uncertainties related to phenology models, optimization algorithms, sampling procedures, and sample sizes, evaluating their effects on model performance and model projection separately in site- and species-specific calibration mode. To this end, we conducted an

extensive computer experiment with 21 autumn phenology models from the literature, 5 optimization algorithms, each run with 2 different settings, and various samples based on random, structured, and stratified sampling procedures and on different sample sizes. We analyzed the performance of >2.3 million combinations of model, algorithm, sample, and calibration mode based on observations for beech, pedunculate oak, and larch from Central Europe for the years 1948–2015 (500 sites per species; PEP725; Templ et al., 2018). Further, we analyzed 39 million projections to the year 2099 according to these combinations under 26 different climate model chains, which were split between 2 different representative concentration pathways (CORDEX EUR-11; RCP 4.5 and RCP 8.5; Riahi et al., 2011; Thomson et al., 2011; Jacob et al., 2014). We addressed the following research questions:

I.     What is the effect of the phenology model and calibration approach (i.e., calibration mode, optimization algorithm, and calibration sample) on model performance and projections?

II.    What is the effect of sample size on the degree to which models are overfitted or represent the entire population?

III.   Do better performing models lead to more accurate predictions?

## 2     Data and methods

### 2.1     Data

#### 2.1.1     Phenological observations

We ran our computer experiment with leaf phenology observations from Central Europe for common beech (*Fagus sylvatica* L.), pedunculate oak (*Quercus robur* L.), and European larch (*Larix decidua* MILL.). All phenological data were derived from the PEP725 project database (http://www.pep725.eu/; accessed on April 13, 2022). The PEP725 dataset mainly comprises data from 1948–2015 that were predominantly collected in Austria, Belgium, Czech Republic, Germany, the Netherlands, Switzerland, and the United Kingdom (Templ et al., 2018). We only considered site-years for which the phenological data were in the proper order (i.e. the first leaves have separated before they unfolded, BBCH10 before BBCH11, and 40% of the leaves have colored or fallen before 50% of the leaves, BBCH94 before BBCH95; Hack et al., 1992; Meier, 2001) and the period between spring and autumn phenology was at least 30 days. Subsequently, we only considered sites with at least 20 years for which both spring and autumn phenology data were available. We randomly selected 500 of these sites per species. Each of these sites comprised 20–65 (beech), 20–64 (oak), or 20–30 (larch) site-years, all of which included a datum for spring and autumn phenology. This added up to 17 211 site-years for beech, 16 954 site-years for oak, and 11 602 site-years for larch. Spring phenology corresponded to BBCH11 for beech and oak and BBCH10 for larch, while autumn phenology for all three species was represented by BBCH94, hence forward referred to as leaf coloration (Hack et al., 1992; Meier, 2001).

The 500 selected sites per species differed in location as well as in leaf phenology and climatic conditions. Most sites were from Germany but also from other countries such as Slovakia or Norway (Fig. 1). Autumn phenology averaged over

selected site-years per site ranged from day of year 254–308 (beech), 265–309 (oak), and 261–314 (larch). Corresponding average mean annual temperatures ranged from 0.6–11.0 °C (beech), 6.3–11.0 °C (oak), and 4.1–11.0 °C (larch) and annual precipitation ranged from 470–1272 mm (beech), 456–1232 mm (oak), and 487–1229 mm (larch; Fig. 1).

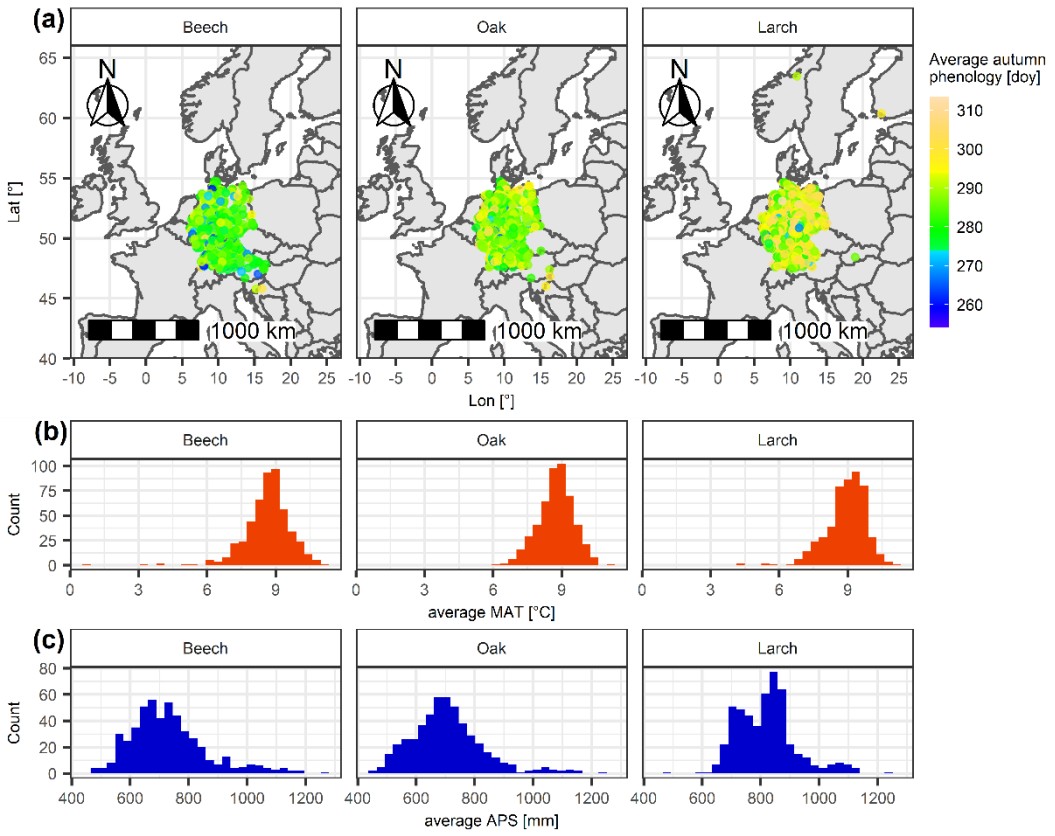

**Figure 1: Sites of considered leaf phenology data with respective average climatic conditions for beech, oak, and larch.** (a) The location of each site is marked with a dot, the color of which indicates the average day of year of autumn phenology. (b) and (c) show the distribution of average mean annual temperature (MAT; [°C]) and average annual precipitation sum (APS; [mm]) per site.

### 2.1.2    Model drivers

The daily and seasonal drivers of the phenology models were derived and calculated from interpolated daily weather data as well as data of different time scales of short- and longwave radiation, atmospheric $CO_2$ concentration, leaf area indices, and soil moisture. Daily drivers are daily minimum air temperature which is mostly combined with day length (cf. Sect. 2.2). Some models further consider seasonal drivers, which we derived from daily mean and maximum air temperature, precipitation, soil moisture, net and downwelling shortwave radiation, and net longwave radiation, from monthly leaf area indices, from monthly or yearly atmospheric $CO_2$ concentration data, as well as from site-specific plant-available water capacity data. We calculated day length according to latitude and day of year (Supplement S3: Eq. S1; Brock, 1981). The other daily variables were derived

from two NASA global land data assimilation system datasets on a $0.25° \times 0.25°$ grid (~25 km; GLDAS-2.0 and GLDAS-2.1 for the years 1948–2000 and 2001–2015, respectively; Rodell et al., 2004; Beaudoing and Rodell, 2019, 2020) for the past and from the CMIP5-based CORDEX-EUR-11 datasets on a rotated $0.11° \times 0.11°$ grid (~12.5 km; for the years 2006–2099; Riahi et al., 2011; Thomson et al., 2011; Jacob et al., 2014) for two representative concentration pathways (RCP). After examining the climate projection data (Supplement S1: Sect. 2), we remained with 16 and 10 climate model chains (CMC; i.e. particular combinations of global and regional climate models) for the RCP 4.5 and 8.5, respectively. Atmospheric $CO_2$ concentrations were derived from the historical CMIP6 and observational Mauna Loa datasets for the years 1948–2014 and for 2015, respectively (monthly data; Meinshausen et al., 2017; Thoning et al., 2021) for the past and from the CMIP5 datasets for the years 2006–2099 (yearly data; Meinshausen et al., 2011) for the climate projections, matching the RCP 4.5 and RCP 8.5 scenarios (Smith and Wigley, 2006; Clarke et al., 2007; Wise et al., 2009; Riahi et al., 2007). Leaf area indices were derived from the GIMMS LAI3g dataset on a $0.25° \times 0.25°$ grid, averaged over the years 1981–2015 (Zhu et al., 2013; Mao and Yan, 2019). The plant-available water capacity per site was derived directly or estimated according to soil composition (i.e., volumetric silt, sand, and clay contents) from corresponding ISRIC SoilGrids250m datasets on a $250\,m \times 250\,m$ grid (versions 2017-03 or 2.0 for water content or soil composition, respectively; Hengl et al., 2017). More detailed information about the applied driver data and driver calculation is given in Supplement S1 and S3, respectively.

## 2.2    Phenology models

We based our analysis on 21 process-oriented models of autumn phenology, which differ in their underlying functions and the drivers they consider (Table 1; Meier, 2022). In all models, the projected date for autumn phenology corresponds to the first day of the year, for which an accumulated daily senescence rate ($R_S$) exceeds a corresponding threshold value. $R_S$ responds to daily minimum temperature (cf. Sect 4.5.2) and, except for the CDD model, to day length (see Table 1 and Supplement S2). While the senescence rate increases with cooler temperatures, it may increase or decrease with shorter days, depending on the response function. Thus, with cooler temperatures, the rate follows either a monotonically increasing response curve (Mon; with $R_S \geq 0$) or a sigmoidal response curve (Sig; with $0 \leq R_S \leq 1$), with the monotonous increase weakened or amplified with shorter days (Mon– or Mon+), depending on the model (Dufrêne et al., 2005; Delpierre et al., 2009; Lang et al., 2019). The threshold value for the accumulated rate is either a constant (Co) or depends linearly on one or two seasonal drivers (Li). Accumulation of the daily rate starts on the first day after the 173rd day of the year (summer solstice) or after the 200th day of the year, for which minimum temperature and/or day length fall below corresponding thresholds. The models have between 2 and 7 free parameters, which are jointly fitted during calibration.

**Table 1. Compared process-oriented models of autumn phenology grouped according to their response curve for the daily senescence rate and their corresponding threshold function.**

| Response curve (threshold function) | Model | Daily drivers | Seasonal drivers | Number of free parameters | Source |
|---|---|---|---|---|---|
| Mon (Co) | CDD | T | - | 2 | Du05 |
| Mon− (Co) | DM1 | T, L | - | 5 | De09 |
| | DM1$_{Za20}$ | | | 3 | Za20 |
| Mon− (Li) | SIAM | T, L | a.dSP | 4 | Ke15 |
| | TDM1 | | T$_{LS}$ | 6 | Li19 |
| | PDM1 | | LPI$_{LS}$ | 6 | Li19 |
| | TPDM1 | | T$_{LS}$, LPI$_{LS}$ | 7 | Li19 |
| Mon+ (Co) | DM2 | T, L | - | 5 | De09 |
| | DM2$_{Za20}$ | | | 3 | Za20 |
| Mon+ (Li) | TDM2 | T, L | T$_{LS}$ | 6 | Li19 |
| | PDM2 | | LPI$_{LS}$ | 6 | Li19 |
| | TPDM2 | | T$_{LS}$, LPI$_{LS}$ | 7 | Li19 |
| Sig (Co) | TPMt | T, L | - | 4 | La19 |
| | TPMp | | | 4 | La19 |
| Sig (Li) | SIAM$_{Za20}$ | T, L | a.dSP | 5 | Za20 |
| | TDM$_{Za20}$ | | T$_{GS}$ | 5 | Za20 |
| | PDM$_{Za20}$ | | LPI$_{Za20}$ | 5 | Za20 |
| | TPDM$_{Za20}$ | | T$_{GS}$, LPI$_{Za20}$ | 6 | Za20 |
| | PIA$_{GSI}$ | | a.GSI | 5 | Za20 |
| | PIA$^{+}$ | | a.A$_{net}$ | 5 | Za20 |
| | PIA$^{-}$ | | a.A$_{net-w}$ | 5 | Za20 |

Note: Daily senescence rate responds to the daily drivers minimum temperature (T) and day length (L), following either a monotonically increasing curve (Mon) with cooler temperatures, which may be weakened or amplified with shorter days (Mon− or Mon+), or a sigmoidal curve (Sig). The threshold value is either a constant (Co) or a linear function (Li) of one or two of the following seasonal drivers: site-specific anomaly of (1) spring phenology (a.dSP), (2) growing season index (a.GSI), and (3) daytime net photosynthesis accumulated during the growing season ignoring or considering water limitation constraints (a.A$_{net}$ and a.A$_{net-w}$), as well as the actual (4) leafy season or growing season mean temperature (T$_{LS}$ and T$_{GS}$), (5) low precipitation index averaged over the leafy season (LPI$_{LS}$), or (6) adapted low precipitation index of the growing season (LPI$_{Za20}$). Further, the number of free parameters fitted during model calibration and the sources for each model are listed (i.e., De09: Delpierre et al. (2009); Du05: Dufrêne et al. (2005); Ke15: Keenan and Richardson (2015); La19: Lang et al. (2019); Li19: Liu et al. (2019); Za20: Zani et al. (2020)). Note that the models CDD, DM1, DM2, SIAM, TDM1, TDM2, PDM1, PDM2, TPDM1, and TPDM2 are originally driven by daily mean rather than daily minimum temperature (cf. Sect. 4.5.2). All models are explained in detail in Supplement S2.

While all models differ in their functions and drivers considered, they can be grouped according to the formulation of the response curve of the senescence rate and of the threshold function (Table 1). Models within a particular group differ by the number of free parameters, by the determination of the initial day of the accumulation of the senescence rate, or by the seasonal drivers of the threshold. The difference in the number of free parameters is relevant for the groups Mon− (Co) and Mon+ (Co). These groups contain two models each, which differ by the two exponents for the effects of cooler and shorter days on the senescence rate. Each of these exponents can be calibrated to the values 0, 1, or 2 in the models with more parameters, whereas the exponents are set to 1 in the models with fewer parameters. The initial day of the accumulation of the senescence rate is either defined according to temperature or day length in the two models of the group Sig (Co). The one or two seasonal drivers considered by the models of the groups Mon− (Li), Mon+ (Li), and Sig (Li) are site-specific anomalies of the timing of spring phenology, the growing season index, and daytime net photosynthesis accumulated during the growing season ignoring or considering water limitation constraints, as well as the actual leafy season or growing season mean temperature, the low precipitation index averaged over the leafy season, or the adapted low precipitation index of the growing season. All models are explained in detail in Supplement S2).

## 2.3 Model calibration and validation

### 2.3.1 Calibration modes

We based our study on both a site- and species-specific calibration mode. In the site-specific mode, we derived for every calibration a species- and site-specific set of parameters (i.e. every combination of optimization algorithm and sample). In the species-specific mode, we derived for every calibration a species-specific set of parameters based on the observations from more than one site, depending on the calibration sample. Model performances were estimated with an external model validation, namely with a 5-fold cross-validation (James et al., 2017, Ch. 5.1.3) and a separate validation sample in the site- and species-specific mode, respectively.

### 2.3.2 Optimization algorithms

We calibrated the models with five different optimization algorithms, which can be grouped into Bayesian and non-Bayesian algorithms. The two Bayesian algorithms that we evaluated are Efficient Global Optimization algorithms based on kriging (Krige, 1951; Picheny and Ginsbourger, 2014): one is purely Bayesian (EGO), whereas the other combines Bayesian optimization with a deterministic trust-region formation (TREGO). The three non-Bayesian algorithms that we evaluated are Generalized Simulated Annealing (GenSA; Xiang et al., 1997; Xiang et al., 2013), Particle Swarm Optimization (PSO; Clerc, 2011, 2012; Marini and Walczak, 2015), and Covariance Matrix Adaptation with Evolutionary Strategies (CMA-ES; Hansen, 2006, 2016). Every Bayesian and non-Bayesian algorithm was executed in a normal and extended optimization mode, i.e. with few and many iterations/steps (norm. and extd., respectively; Supplement S4: Table S1). In addition, the parameter boundaries within which all algorithms searched for the global optimum (Supplement S2: Table S1) were scaled to range from 0 to 1. All

algorithms optimized the free model parameters to obtain the smallest possible root mean square error (RMSE; Supplement S4: Eq. S1) between the observed and modelled days of year of autumn phenology. As the Bayesian algorithms cannot handle iterations that produce NA values (i.e. modelled day of year > 366), such values were set to day of year = 1 for all algorithms and before RMSE calculation.

### 2.3.3 Calibration and validation samples

Calibration and validation samples can be selected according to different sampling procedures with different bases (e.g., randomly or systematically based on the year of observation) and have different sizes (i.e. number of observations and/or number of sites). Here, we distinguished between the sampling procedures random, systematically continuous, systematically balanced, and stratified. Further, our populations consisted of sites that included between 20 and 65 years, which directly affected the sample size in the site-specific calibration mode. In the species-specific mode, we calibrated the models with samples that ranged from 2 to 500 sites.

In the site-specific mode, the observations for the 5-fold cross-validation were selected (1) randomly or (2) systematically (Supplement S4: Fig. S1). For the random sampling procedure, the observations were randomly assigned to one of five validation bins. For the systematic sampling procedure, we ranked the observations based on the year, mean annual temperature (MAT), or autumn phenology date (AP) and created five equally sized samples containing continuous or balanced (i.e. every 5th) observations (see Supplement S4: Sect. 2.1 for further details regarding these procedures). Hence, every model was calibrated seven times for each of the 500 sites per species, namely with a randomized or a time-, phenology-, or temperature-based systematically continuous or systematically balanced cross-validation. This amounted to 2 205 000 calibration runs (i.e. 500 sites × 3 species × 21 models × 5 optimization algorithms × 2 optimization modes × 7 sample selection procedures) that consisted of 5 cross-validation runs each. Further, for the projections, every model was calibrated with all observations per site and species.

In the species-specific mode, we put aside 25% randomly selected observations per site and per species (rounded up to the next integer) for external validation samples and created various calibration samples from the remaining observations, selecting the different sites with different procedures. These calibration samples either contained the remaining observations of all 500 sites (full sample) or of (1) randomly selected, (2) systematically selected, or (3) stratified sites per species (Supplement S4: Fig. S2). The random and systematic samples contained the observations of 2, 5, 10, 20, 50, 100, or 200 sites. Randomly sampled sites were chosen either from the entire or half the population, with the latter being determined according to MAT and AP (i.e. cooler average MAT or earlier or later average AP). The systematically sampled sites were selected according to a balanced procedure in which the greatest possible distance between sites ranked by average MAT or AP was chosen. (Note that the distance between the first and last site was 490, not 500 sites, allowing up to ten draws with a parallel shift of the first and last site.) The stratified samples consisted of one randomly drawn site from each of 12 MAT- or 17 AP-based bins. The chosen bin widths maximized the number of equal-sized bins so that they still contained at least one site (see Supplement S4: Sect. 2.2 for further details regarding these procedures). We drew five samples per procedure and size, except

for the full sample, which we drew only once, as it contained fixed sites, namely all sites in the population. Altogether, this amounted to 139 230 calibration runs (i.e. 3 species × 21 models × 5 optimization algorithms × 2 optimization modes × (6 sample selection procedures × 7 sample sizes × 5 draws + 2 sample selection procedures × 5 draws + 1 sample selection procedure)) that differed in the size and selection procedure of the corresponding sample. Every calibration run was validated with the sample-specific and population-specific external validation sample. While the former consisted of the same sites as the calibration sample, the latter consisted of all 500 sites and hence was the same for every calibration run per species. Every calibration run was validated with the sample-specific and population-specific external validation sample, hence forward referred to as "validation within sample" and "validation within population". While the former consisted of the same sites as the calibration sample, the latter consisted of all 500 sites and hence was the same for every calibration run per species.

## 2.4    Model projections

We projected autumn phenology to the years 2080–2099 for every combination of phenology model, calibration mode, optimization algorithm, and calibration sample that converged without producing NA values, assuming a linear trend for spring phenology. While non-converging runs did not produce calibrated model parameters, we further excluded the converging runs that resulted in NA values in either the calibration or validation. In addition, we excluded combinations where projected autumn phenology occurred before the 173rd or 200th day of the year (i.e. the earliest possible model-specific day of senescence rate accumulation). Thus, we received 41 901 704 site-specific time series for the years 2080–2099 of autumn phenology projected with site-specific models, hereafter referred to as site-specific projections. These time series differed in climate projection scenario (i.e. per combination of representative concentration pathway and climate model chain), phenology model, optimization algorithm, and calibration sample. Species-specific models led to projections for all 500 sites per species (i.e. the entire populations) and thus to 1 574 378 000 time series for the years 2080–2099 that differed in climate projection scenario, model, algorithm, and calibration sample, hereafter referred to as species-specific projections. For site- and species-specific models, we projected the spring phenology relevant for the seasonal drivers assuming a linear trend of −2 days per decade (Piao et al., 2019; Menzel et al., 2020; Meier et al., 2021). This trend was applied from the year after the last observation (ranging from 1969 to 2015, depending on site and species) and was based on the respective site average over the last 10 observations per species.

## 2.5    Proxies and statistics

### 2.5.1    Sample size proxies

We approximated the effect of sample size (1) on the bias-variance trade-off and (2) on the degree to which models represent the entire population with the respective size proxies (1) number of observations per parameter and (2) site ratio. The effect of sample size on the bias-variance trade-off may depend on the number of observations in the calibration sample ($N$) relative to the number of free parameters ($q$) in the phenology model. In other words, a sample of, say, 50 observations may lead to a

better calibration of the CDD model (2 free parameters) compared to the TPDM1 model (7 free parameters). In the site-specific calibration, we calculated for each site and model the ratio N:q with $N$ being 80% of the total number of observations per site to account for the 5-fold cross-validation. Assuming the 50 observations in the example above are the basis for the 5-fold cross-validation, $N$ becomes 40, resulting in N:q = 40/2 for the CDD model and N:q = 40/7 for the TPDM1 model. With species-specific calibration, we considered the *average* number of observations per site $(\overline{N})$ and calculated for each calibration sample and model the ratio $\overline{N}$:q to separate this ratio from the site ratio explained further below. Assuming the 50 observations in the example above correspond to a calibration sample based on 2 sites, $\overline{N}$ becomes 25, resulting in $\overline{N}$:q = 25/2 for the CDD model and $\overline{N}$:q = 25/7 for the TPDM1 model. The effect of sample size on the degree to which models represent the entire population with species-specific calibration may depend on the number of sites in the calibration sample ($s$) relative to the number of sites in the entire population ($S$; i.e. site ratio; s:S). Thus, we derived the site ratio by dividing $s$ by 500. Note that the combined ratios $\overline{N}$:q and s:S account for the effect of the total sample size as $(\overline{N} \times s) / (q \times S) = N / (q \times S)$.

## 2.5.2    Model performance

We quantified the performance of each calibrated model according to the root mean square error (RMSE; Supplement S4: Eq. S1). The RMSE was calculated for the calibration and the validation samples (i.e. internal and external RMSE, respectively) as well as at the sample- and population level with the species-specific calibration (i.e. validated within sample or population; external sample RMSE or external population RMSE, respectively). We derived each RMSE per sample at the site level with the site-specific calibration and at the sample level with the species-specific calibration.

To measure the effect (1) on the bias-variance trade-off and (2) on the degree to which models represent the entire population, we derived two respective RMSE ratios. Regarding the bias-variance trade-off and with the site-specific calibration, we divided the external RMSE by the internal RMSE derived from the calibration run with all observations per site and species (Cawley and Talbot, 2010, Sect. 5.2.1; James et al., 2017, Ch. 2.2.2). The numerator was expected to be larger than the denominator and increasing ratios were associated with an increasing bias, indicating overfitting. Regarding the degree to which models represent the entire population and hence with the species-specific calibration, we divided the external sample RMSE by the external population RMSE. Here, the numerator was expected to be smaller than the denominator and increasing ratios were associated with increasing representativeness.

We applied two different treatments to calibration runs that led to NA values (i.e. the threshold value was not reached by the accumulated senescence rate until day of year 366) or did not converge at all (Supplement S6: Fig. S1). On the one hand, the exclusion of such runs may bias the results regarding model performance, since a non-converging model must certainly be considered to perform worse than a converging one. Therefore, in contrast to the model calibration, we replaced the NA values with the respective observed date + 170 days (i.e. a difference that exceeds the largest modelled differences in any calibration or validation sample by two days) and assigned an RMSE of 170 to non-converging runs before we analyzed the model performance. On the other hand, replacing NA values with a fixed value leads to an artificial effect and affects the performance analysis, as, say, a linear dependence of the RMSE on a predictor is suddenly interrupted. Moreover, projections

based on models that converged but produced NA values seem questionable, while projections based on non-converging models are impossible. Therefore, we implemented a second analysis of performance, from which we excluded the calibration runs that did not converge or contained one or more NA values in either the calibration or the validation sample. Our main results regarding model performance were based on the substituted NA values and the RMSE of 170 days for non-converging runs. However, where necessary, we have referred to the results based only on converged runs without NA values (provided in Supplement S6: Sects. 2.1.2 and 2.2.2). Furthermore, our results regarding projections and our comparisons between performance and projections are based solely on converging runs without NA values.

### 2.5.3    Model projections

We analyzed the site- and species-specific projections of autumn phenology according to a 100-year shift ($\Delta_{100}$) at the site level. $\Delta_{100}$ was defined as the difference between the means of the observations for the years 1980–1999 and of the projections for the years 2080–2099. If observations for the years 1980–1999 were missing, we used the mean of the 20 last observations instead. Thus, the derived shift was linearly adjusted to correspond to 100 years.

### 2.6    Evaluation of model performance and autumn phenology projections

To answer research question I (RQ I), we calculated the mean, median, standard deviation, and skewness of the RMSE distributed across phenology models, optimization algorithms, sampling procedures, and (binned) sample size proxies. These statistics were derived separately per site- and species-specific calibration validated within sample or population, giving a first impression of the effects on model performance. Further, the distribution of the RMSE was relevant for subsequent evaluations.

To answer RQ I and RQ II, we estimated the effects of phenology models, optimization algorithms, sampling procedures, and sample size proxies on model performance and, together with climate projection scenarios, on model projections with generalized additive models (GAMs; Wood, 2017) and subsequent analyses of variance (ANOVA; Fig. 2; Chandler and Scott, 2011). We fitted the GAMs separately per calibration and projection mode, i.e. per site- and species-specific calibration validated (projected) within sample or population (Supplement S5: Sect. 1; Supplement S5: Eqs. S1 and S2). The response variables RMSE and $\Delta_{100}$ were assumed to depend linearly on the explanatory factors phenology models, optimization algorithms, sampling procedures, and climate projection scenarios (only regarding $\Delta_{100}$) as well as on the continuous sample size proxies as explanatory variable. Sites and species were included as smooth terms, which were set to crossed random effects such that the GAM mimicked linear mixed-effects models with random intercepts (Supplement S5: Sect. 2; Pinheiro and Bates, 2000; see the discussion for reasons and implications of our choice). The RMSE and sample size proxies were log-transformed, since they can only take positive values and followed right-skewed distributions (i.e. skew $> 0$; Supplement S6: Tables S1–S4). Coefficients were estimated with fast restricted maximum likelihood (Wood, 2011). Thereafter, the fitted GAMs served as input for corresponding type-III ANOVA (Supplement S5: Sect. 4; Yates, 1934; Herr, 1986; Chandler and Scott, 2011), with which we estimated the influence on the RMSE and the $\Delta_{100}$. The influence was expressed as the relative variance in RMSE or in $\Delta_{100}$ explained by phenology models, optimization algorithms, sampling

procedures, sample size proxies, and climate projection scenarios (only regarding $\Delta_{100}$). Regarding model projections, we drew five random samples of $10^5$ projections per climate projection scenario and per $\Delta_{100}$ projected with site- and species-specific models within sample or population. Thereafter, we fitted a separate GAM and ANOVA for each of these 15 samples (see the discussion for reasons and implications of this approach).

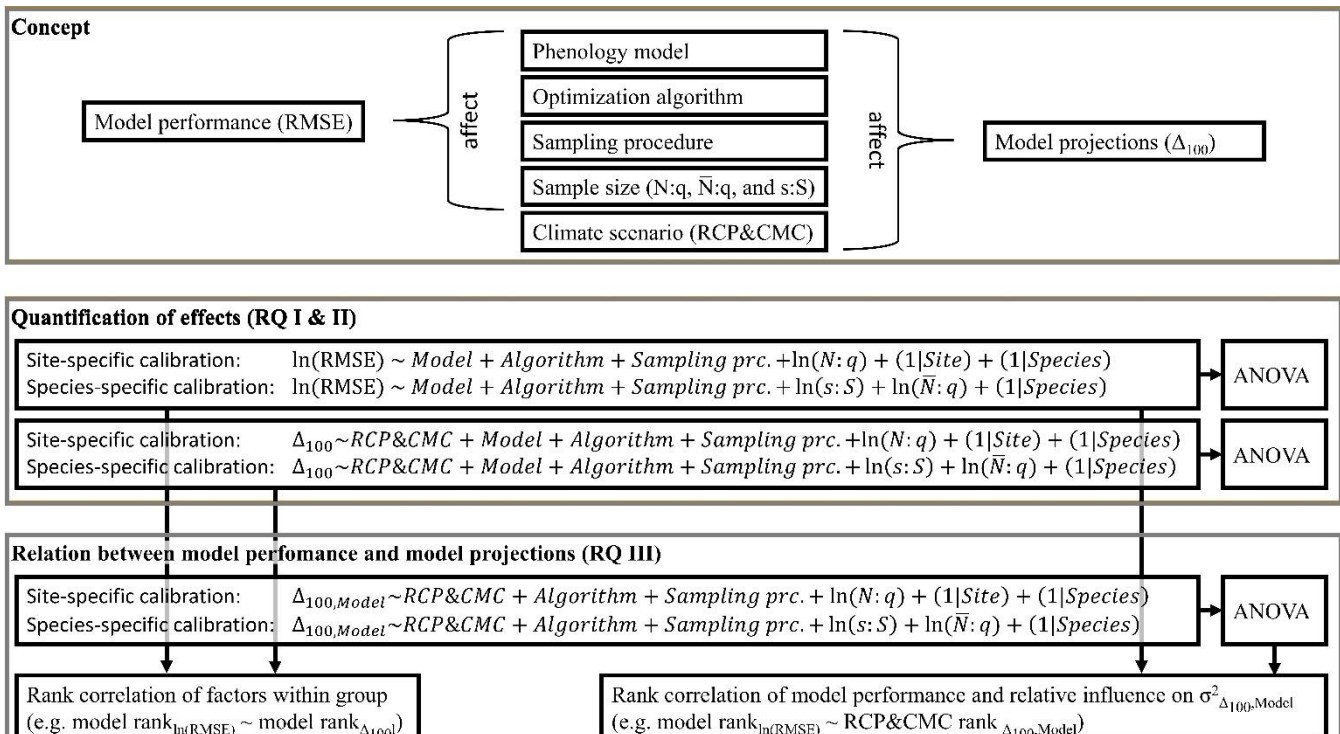

**Figure 2: Concept and methods applied.** This study assumed that, in addition to phenology models and climate scenarios, the choice of optimization algorithm and calibration sample (i.e. sampling procedure and sample size) affect model performance and model projections (i.e. the root mean square error, RMSE, and the shift between autumn leaf phenology in 2080–2099 and 1980–1999, $\Delta_{100}$, respectively). To answer research questions I and II (RQ I & II), the effects of these factors on the RMSE and $\Delta_{100}$ were quantified with linear mixed-effects models. Subsequently, the relative influence of the factors (e.g. all phenology models) on the explained variance ($\sigma^2$) of RMSE and $\Delta_{100}$ were quantified with type-III ANOVA. To answer RQ III, the effects on the RMSE were related to the effects on $\Delta_{100}$ and the influences on $\sigma^2_{\Delta_{100}}$ by calculating the Kendall rank correlations (e.g., between the effects of the phenology models on the RMSE and $\Delta_{100}$ or between the effect of the phenology models on the RMSE and the influence of each model on $\sigma^2_{\Delta_{100}}$). The phenology models were calibrated site- and species-specifically (i.e. one set of parameters per site and species vs. one set of parameters per species, respectively). Sample size was quantified by the number of observations relative to the number of free parameters in the phenology model (N:q), the average number of observations relative to the number of free parameters ($\overline{N}$:q), and the number of sites relative to the 500 sites of the entire population (s:S).

The coefficient estimates of the GAMs expressed (relative) changes towards the corresponding reference levels of RMSE or $\Delta_{100}$. The reference levels were based on the CDD model calibrated with the GenSA (norm.) algorithm on a randomly selected sample and, only regarding $\Delta_{100}$, projected with the CMC 1 of the RCP 4.5. Hence, the estimates of the intercept refer to the estimated log-transformed RMSE or $\Delta_{100}$ according to these reference levels with sample size proxies of 1 (i.e. log-

transformed proxies of 0). Regarding interpretation of model performance, negative coefficient estimates indicated better performance, which was reflected in smaller RMSE values. The effects of the explanatory variables were expressed as relative changes of the reference RMSE, due to the log-transformation of the latter (Supplement S5; Sect. 3). Regarding model projections, while negative coefficient estimates in combination with negative reference $\Delta_{100}$ resulted in an accelerated projected advancement of autumn phenology, they weakened a projected delay of autumn phenology or changed it to a projected advancement in combination with positive reference $\Delta_{100}$ and vice versa. In other words, negative coefficient estimates only translated into earlier projected autumn phenology when the corresponding reference $\Delta_{100}$ was negative or their absolute values were larger than the (positive) reference $\Delta_{100}$ and vice versa.

To answer RQ III, we related both (1) the ranked effects on model performance to the ranked effects on model projections and (2) the performance ranks of phenology models to the ranked influence of explanatory variables on $\Delta_{100}$ per model (Fig. 2). First, we ranked phenology models, optimization algorithms, and sampling procedures according to their estimated effects on log-transformed RMSE or $\Delta_{100}$, and calculated the Kendall rank correlation (Kendall, 1938) within each group of factors (e.g. within phenology models). Negative correlations indicated, for example, that models with better performance projected later autumn phenology than models with poorer performance and vice versa. Second, we fitted GAMs per site- and per species-specific phenology model to the response variable $\Delta_{100}$ as described above (Supplement S5: Eq. S1) but excluded phenology models from the explanatory variables. We derived type-III ANOVA per GAM (Supplement S5, Eq. S14) and ranked the influence of optimization algorithms, sampling procedures, sample size proxies, and climate projection scenarios across phenology models. Thus, we calculated the Kendall rank correlation (Kendall, 1938) between these newly derived ranks and the ranks of the phenology models based on their effect on performance. In other words, we analyzed if the ranked influence of, for example, aggregated climate projection scenarios on $\Delta_{100}$ correlated with the ranked performance of phenology models. In this example, negative correlations indicated that climate projection scenarios had a larger relative influence on projected autumn phenology when combined with phenology models that performed better than with models that performed worse and vice versa. As before, each GAM and ANOVA was fitted and derived five times per phenology model and climate projection scenario based on five random samples of $10^5$ corresponding projections per climate projection scenario. Therefore, the ranks of optimization algorithms, sampling procedures, sample size proxies, and climate projection scenarios were based on the mean coefficient estimates or mean relative explained variance.

We chose a low significance level and specified Bayes factors to account for the large GAMs with many explanatory variables and the frequent mis- or overinterpretation of $p$-values (Benjamin and Berger, 2019; Goodman, 2008; Ioannidis, 2005; Wasserstein et al., 2019; Nuzzo, 2015). We applied a lower than usual significance level, namely $\alpha = 0.01$ (i.e. p < 0.005 for two-sided distributions; Benjamin and Berger, 2019), and included the 99% confidence intervals in our results. In addition, we complemented the $p$-values with Bayes factors (BF) to express the degree to which our data changed the odds between the respective null hypothesis H0 and the alternative hypothesis H1 (BF01; Johnson, 2005; Held and Ott, 2018). For example, if we assume a prior probability of 20% for the alternative hypothesis (i.e. a prior odds ratio H0:H1 of 80/20 = 4/1), then a $BF_{01}$ of 1/20 means that the new data suggests a posterior odds ratio of 1/5 (i.e. 4/1 × 1/20) and thus a posterior probability of 83.3%

for the alternative hypothesis. Our study was exploratory in nature (Held and Ott, 2018, Sect. 1.3.2), hence our null hypothesis was that there is no effect as opposed to the alternative hypothesis that there is one, for which a local distribution around zero is assumed (Held and Ott, 2018, Sect. 2.2). We derived the corresponding sample-size adjusted *minimum* $BF_{01}$ ($\underline{BF}_{01}$) from the *p*-values of the GAM coefficients, ANOVA, and Kendall rank correlations (Johnson, 2005, Eq. 8; Held and Ott, 2016, Sect. 3; 2018, Sect. 3). While $BF_{01}$ never exceed the value of 1, $BF_{01}$ below 1/100 may be considered "very strong" and $BF_{01}$ above 1/3 may be (very) "weak" (Held and Ott, 2018, Table 2). Hence forward, we refer to results with $p < 0.005$ as significant and with $\underline{BF}_{01} < 1/1000$ as "decisive". Note that the $\underline{BF}_{01}$ expresses the most optimistic shift towards the alternative hypothesis.

All computations for data preparations, calculations, and visualizations were conducted in R (versions 4.0.2 and 4.1.3 for scientific computing and data visualisations, respectively; R Core Team, 2022) with different packages: Data were prepared with data.table (Dowle and Srinivasan, 2021), phenology models were coded based on phenor (Hufkens et al., 2018) and calibrated with DiceDesign and DiceOptim (for the optimisation algorithms EGO and TREGO; Dupuy et al., 2015; Picheny et al., 2021), GenSA via phenor (GenSA; Xiang et al., 2013; Hufkens et al., 2018), pso (PSO; Bendtsen, 2012), and cmaes (CMA-ES; Trautmann et al., 2011), while the RMSE was calculated with hydroGOF (Zambrano-Bigiarini, 2020). The formulas for the GAMs were translated from linear mixed-effects models with buildmer (Voeten, 2022), GAMs were fitted with mgcv::bam (Wood, 2011, 2017), the corresponding coefficients and *p*-values were extracted with mixedup (Clark, 2022), and sample size-adjusted $\underline{BF}_{01}$ were derived from *p*-values with pCalibrate::tCalibrate (Held and Ott, 2018). Summary statistics, ANOVA, and correlations with respective *p*-values were calculated with stats (R Core Team, 2022). Figures and tables were produced with ggplot2 (Wickham, 2016), ggpubr (Kassambara, 2020), and gtable (Wickham and Pedersen, 2019).

## 3 Results

### 3.1 Model performance

We evaluated 2 217 888 and 139 231 externally validated site- and species-specific calibration runs. Each of these runs represented a unique combination of 21 models, 10 optimization algorithms, 7 calibration samples × 500 sites with the site-specific calibration or 9 calibration samples with the species-specific calibration, and 3 species. All samples for the species-specific calibration were drawn 5 times except for the full sample. From the initial site- and species-specific calibration runs, 136 500 and 7 048 runs, respectively, did not converge, while another 373 126 and 12 524 runs led to NA values in either the internal or external validation (Supplement S6: Fig. S1).

### 3.1.1 Observed effects

Across the phenology models, optimization algorithms, and sampling procedures, the observed distribution of the external root mean square error (RMSE) of the pooled species differed considerably between the calibration and validation modes. Overall, the smallest median RMSEs were similar between the site- and species-specific calibration modes, ranging from 10.1 to 12.4 and 11.7 to 12.6 or 12.4 to 12.9 days in the respective site- and species-specific calibration validated within sample or within

population (Fig. 3, Supplement S6: Table S1). The smallest mean RMSE were considerably larger with the site- than with the species-specific calibration (19.2–52.1 vs. 11.6–23.9 or 12.9–24.4 days; grey dots in Fig. 3). Accordingly, standard deviations were larger with the site- than with the species-specific calibration (28.4–66.5 vs. 3.7–36.6 or 1.2–36.0 days; Fig. 3, Supplement S6: Table S1).

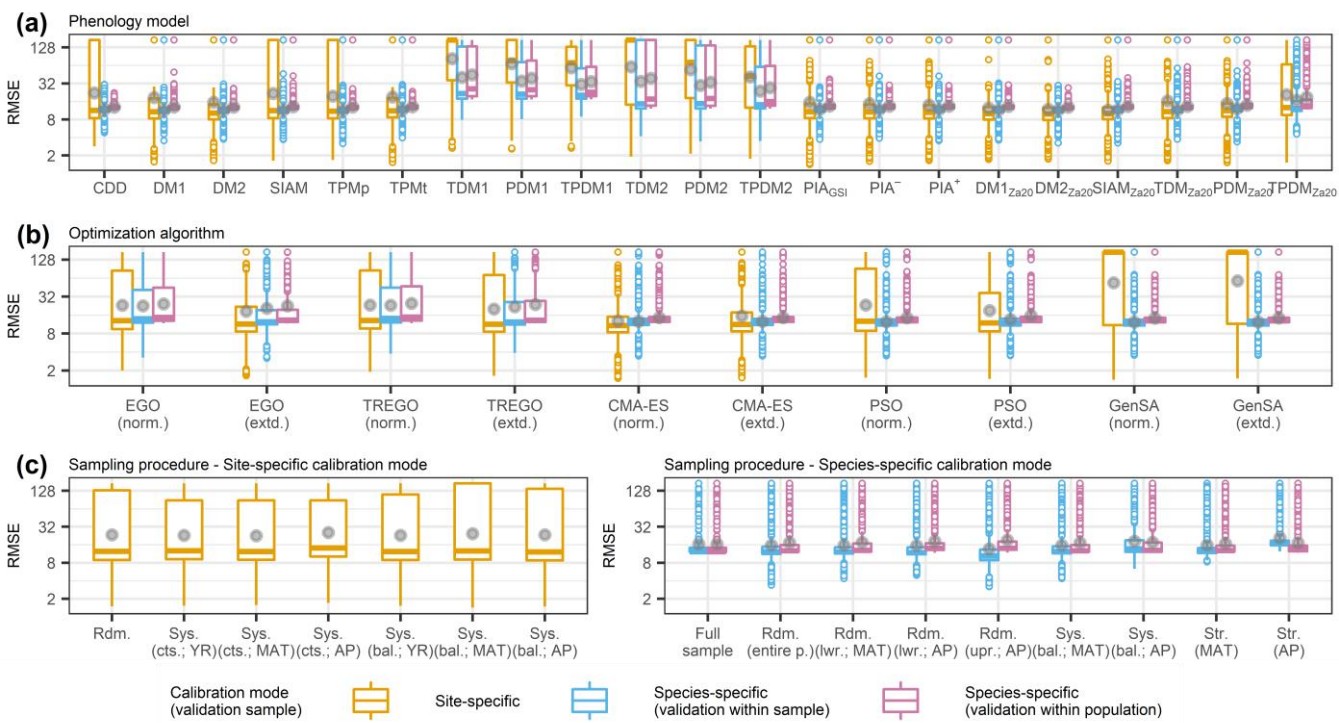

**Figure 3: Observed distributions of the external root mean square error (RMSE) of the pooled species according to (a) phenology models, (b) optimization algorithms, and (c) sampling procedures.** The thick horizontal lines and grey dots indicate the respective median and mean. Boxes cover the inner quartile range, whiskers extend to the most extreme observations or to 1.5 times the inner quartile range, and outliers are indicated as colored circles. Note that the y-axes were log-transformed. In all figures, the colors represent the calibration and validation modes. The abbreviations for the models, algorithms, and sampling procedures are explained in respective Tables Supplement S2: Table S1, Supplement S4: Table S1, and Supplement S4: Table S2/S3.

In the site-specific calibration, increasing sample size relative to the number of free model parameters (N:q) first lowered and then increased the RMSE and generally decreased the bias-variance trade-off. Binned mean RMSE and corresponding standard deviations ranged from 35.8 to 61.1 and from 58.5 to 71.9 days, respectively (Fig. 4c; Supplement S6: Table S2). The binned mean RMSE ratio regarding the bias-variance trade-off (i.e. external RMSE:internal mean RMSE) decreased steadily from 1.5 to 1.0 and at decreasing step sizes with bins of increased N:q. Step sizes between binned N:q were considerably larger for N:q < 9.4 than for N:q ≥ 11.8 (Supplement S6: Table S2), and we observed an abrupt increase in the scatter of the RMSE and RMSE ratio below an N:q of ~9 (Fig. 4c).

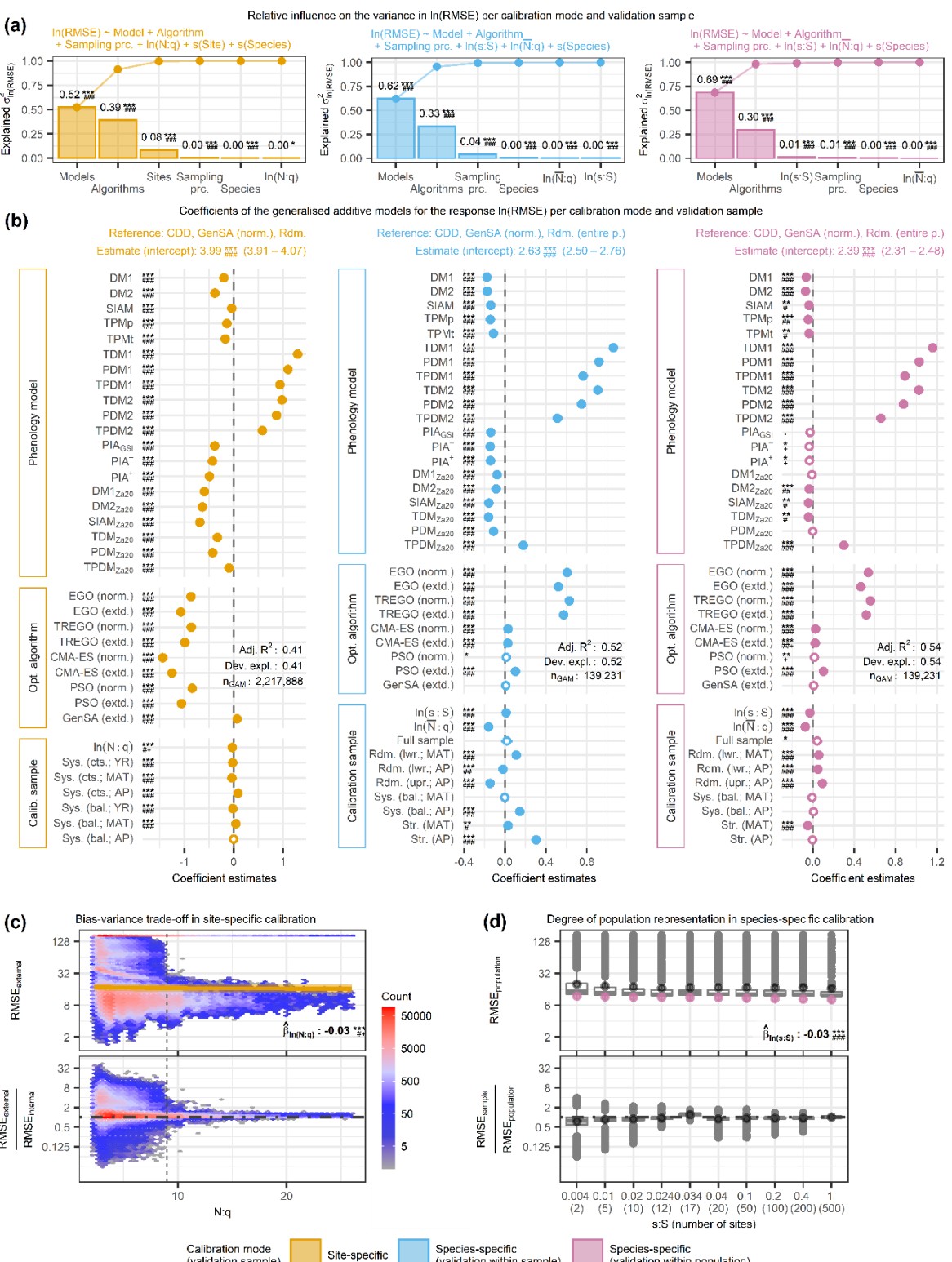

Relative influence on the variance in ln(RMSE) per calibration mode and validation sample

Coefficients of the generalised additive models for the response ln(RMSE) per calibration mode and validation sample

(c) Bias-variance trade-off in site-specific calibration

(d) Degree of population representation in species-specific calibration

Calibration mode (validation sample): Site-specific | Species-specific (validation within sample) | Species-specific (validation within population)

**Figure 4: The relative variance in the log-transformed external root mean square error (RMSE) explained by phenology models, optimization algorithms, and calibration samples (i.e. sampling procedures and sample size proxies) together with the effects of the individual factors and the observed distribution of the RMSE according to sample size proxies.** The relative variance was estimated from analyses of variance (a) based on generalized additive models (GAM; b) for site-specific calibration (left) as well as for within-sample validated (middle) and within-population validated (right) species-specific calibration. Observed distribution is plotted against (c) the number of observations per free model parameter (N:q) and (d) the number of sites relative to the 500 sites of the entire population (s:S), illustrating the bias-variance trade-off with site-specific calibration and the degree to which a model represents the population with species-specific calibration, respectively. In (a), the bars indicate the estimated influence of phenology models, optimization algorithms, sampling procedures, and sample size proxies (N:q, $\overline{N}$:q, and s:S) as well as of the random effects sites and species on the variance in RMSE. The connected dots show the cumulated influence. (b) shows the coefficient estimates (dots or circles) of the GAM together with their 0.5–99.5% confidence limits (whiskers). Dots represent significant ($p < 0.005$) coefficients. Significance levels of each coefficient are indicated with ., *, **, or *** corresponding to $p < 0.1$, 0.05, 0.01, or 0.001, respectively. Further, the minimum Bayes factor is indicated with [+], [#], [#+], [##], [##+], or [###] corresponding to $\underline{BF}_{01} < 1/3$, 1/10, 1/30, 1/100, 1/300, or 1/1000, respectively. Adjusted $R^2$ and deviance explained are printed (*Adj. $R^2$* and *Dev. expl.*, respectively). Note, that negative coefficients (i.e. to the right of the dashed black line) indicate lower RMSE and thus better model performance. In (c), the observed distribution of the actual (top) and relative (bottom) external RMSE are plotted against N:q. The color of the hexagons represents the respective number of observations within the area they cover, and the dashed black line indicates N:q = 9. For the actual RMSE, the estimated effect size according to the GAM is plotted with a solid golden line (median) and a golden shaded area (0.5–99.5% confidence limits). For the relative RMSE, the external RMSE is larger than the internal RMSE in observations above the dot-dashed black line and vice versa. In (d), the observed distribution of the actual (top) and relative (bottom) external population RMSE are plotted for each s:S. For the actual RMSE, the estimated effect size according to the GAM is plotted with purple dots (median) and purple whiskers (0.5–99.5% confidence limits). For the relative RMSE, the external population RMSE is larger than the external sample RMSE in observations above the dot-dashed black line and vice versa. In (c) and (d), the printed values of the corresponding coefficient estimates ($\hat{\beta}$) refer to the log-transformed RMSE and respective N:q or s:S. The abbreviations for the models, algorithms, and sampling procedures in all figures are explained in respective Tables Supplement S2: Table S1, Supplement S4: Table S1, and Supplement S4: Table S2/S3.

In species-specific calibration, larger sample sizes generally co-occurred with smaller population RMSE and higher degrees to which a model represented the population, except for the stratified samples, which led to the best modelled phenology at the population level and the highest degree of population representation. The mean population RMSE and corresponding standard deviation ranged from 24.4 to 30.2 and 36.0 to 40.4 days (Fig. 4d; Supplement S6: Table S2). We observed the smallest mean population RMSE in the stratified sample based on average mean annual temperature (MAT; 12 sites; RMSE = 24.4 days [d]), followed by the stratified sample based on average autumn phenology (AP; 17 sites; RMSE = 25.9 d), and the full sample (500 sites; RMSE = 25.9 d). Except for the stratified samples, increasing sample size resulted in a steady increase in the RMSE ratio regarding the degree to which a model represents the population, which indicated a generally better representation of the population with larger samples. The ratio for stratified MAT samples followed just behind that of the full sample and was 0.95, followed by the samples with 200 and 100 sites that led to a ratio of 0.94. The ratio for the stratified AP samples even exceeded that of the full sample and was 1.21 (i.e. the validation within the samples led to larger RMSE than the validation within the population). Thus, the most accurate modelling of autumn phenology at the population level was achieved with stratified MAT samples rather than with the full sample. Furthermore, models calibrated with AP samples performed better when applied to the whole population, suggesting that the population is better represented with AP samples than with the full sample.

### 3.1.2 Estimated effects

The evidence in the analyzed data against H0 was significant and decisive for the estimated effects ($p < 0.005$ and $\underline{BF}_{01} < 1/1000$) and influences ($p < 0.01$ and $\underline{BF}_{01} < 1/1000$) of most factors, while the deviances explained ranged from 0.41 to 0.67 (Fig. 4, Supplement S6: Fig. S4, Supplement S6: Tables S8–S11 and S15–S18).

Phenology models had generally the largest influence on model performance among optimization algorithms, sampling procedures, sample size, sites, and species, but the degree of influence as well as the best performing models depended on the calibration mode (Fig. 4, Supplement S6: Tables S5–S8). Phenology models explained 52% and 62% or 69% of the variance in RMSE in the site- and species-specific calibration when validated with the sites of the sample or the entire population, respectively (Fig. 4, Supplement S6: Table S8). We estimated the effect on the RMSE of each model relative to the reference CDD model and per calibration mode. In the site-specific calibration, the effects were generally larger than in the species-specific calibration (Fig. 4, Supplement S6: Tables S9–S11). Further, the ranks of the models according to their effects differed between calibration modes (Fig. 4, Supplement S6: Tables S5–S7).

In the site-specific calibration, all 20 phenology models had decisive and significant effects compared to the RMSE of the reference CDD model, which ranged from halving to tripling, and their ranking depended strongly on the treatment of NA values and non-converging runs (Fig. 4 and Supplement S6: Fig. S4, Supplement S6: Tables S5 and S12). The reference model CDD led to an RMSE of 49.8–58.6 days in site-specific calibration (99% confidence interval, $CI_{99}$; see Supplement S5: Sect. 3 for the back-transformation of the coefficient estimates; Fig. 4, Supplement S6: Tables S5 and S9). The largest reduction from this RMSE was achieved with the models $SIAM_{Za20}$, $DM2_{Za20}$, $DM1_{Za20}$, and $PIA^+$, ranging from −50% to −39% ($CI_{99}$). Model ranks and effect sizes changed considerably if NA-producing and non-converging calibration runs were excluded: The RMSE of the reference model CDD dropped to 6.0–7.6 days ($CI_{99}$) and was not reduced by any of the other models (Supplement S6: Fig. S4, Supplement S12: Table S16). In other words, if only calibration runs without NAs were considered, the CDD model performed best, followed by SIAM, $DM2_{Za20}$, and TPMp.

In the species-specific calibration, all 20 or 9 models had decisive and significant effects compared to the reference model CDD if validated within sample or population, respectively, with effects ranging from a reduction by one-fifth to a tripling and resulting in fairly consistent model ranks between the two NA treatments (Fig. 4, Supplement S6: Fig. S4, Supplement S6: Tables S6–S7 and S13–S14). The RMSE according to the reference model CDD was 12.1–15.7 or 10.1–11.9 days ($CI_{99}$) if validated within sample or population, respectively (Fig. 4, Supplement S6: Tables S6–S7 and S10–S11). According to the within-sample validation, the RMSE was reduced the most with the DM1, DM2, $TDM_{Za20}$, and $SIAM_{Za20}$, with reductions between −16% and −15% ($CI_{99}$). According to the within-population validation, the models DM2, DM1, TPMp, and $TDM_{Za20}$ reduced the RMSE the most and reductions ranged from -10% to -1% ($CI_{99}$; Fig. 4, Supplement S6: Tables S6–S7 and S10–S11). If NA-producing and non-converging runs were excluded, the reference RMSE increased to 16.2–20.2 or 12.9–13.9 days ($CI_{99}$; validated within sample or population, respectively), while model ranks were changed in two positions (Supplement S6: Fig. S4, Supplement S6: Tables S13–S14).

Optimization algorithms had the second largest influence on model performance, explaining about one-third of the variance in RMSE, with differences between the calibration modes and NA treatments regarding degree of influence and ranking of individual algorithms (Fig. 4, Supplement S6: Fig. S4, Supplement S6: Tables S5–S8 and S12–S15). Algorithms explained 39% and 33% or 30% of the variance in RMSE in site- and species-specific calibration validated within sample or population, respectively (Fig. 4, Supplement S6: Table S8). In the site-specific calibration, both CMA-ES algorithms (norm. and extd.) resulted in the smallest RMSEs, which were −76% to −71% ($CI_{99}$) lower than the RMSE according to the reference GenSA (norm.; $CI_{99}$ of 49.9–58.6 days, see above; Fig. 4, Supplement S5: Tables S5 and S9). In species-specific calibration, the best results were obtained with both GenSA algorithms (norm. and extd.), whereas the Bayesian algorithms (EGO and TREGO, norm. and extd.) performed worst and resulted in RMSEs +57% to +91% ($CI_{99}$) larger the RMSE according to the reference RMSE ($CI_{99}$ of 12.1–15.7 or 10.0–11.9 days if validated within sample or population, see above; Fig. 4, Supplement S6: Tables S6–S7 and S10–S11). If NA-producing and non-converging calibration runs were excluded, the lowest and largest RMSEs with site-specific calibration were obtained with the GenSA and the Bayesian algorithms, respectively. With species-specific calibration, we observed little change when only calibration runs without NAs were analyzed: As before, Bayesian algorithms led to the largest RMSEs, while both GenSA algorithms resulted in the smallest RMSEs (Supplement S6: Fig. S4, Supplement S6: Tables S12–S14).

Sampling procedures had little influence on model performance in general (third or fourth largest) and were more important with species- than with site-specific calibration where stratified sampling procedures led to the best results (Fig. 4, Supplement S6: Tables S5–S8). Sampling procedures explained 0.3% and 4.0% or 0.7% of the variance in RMSE with site- and species-specific calibration validated within sample or population, respectively (Fig. 4, Supplement S6: Table S8). With site-specific calibration, systematically continuous samples based on mean annual temperature (MAT) and year performed best, diverging by −4.3% to −1.3% ($CI_{99}$, Fig. 4, Supplement S6: Tables S5 and S9) from the RMSE according to the reference random sampling. With species-specific calibration, we received the lowest RMSEs with random samples from half of the population (split according to MAT or autumn phenology, AP) when validated within sample (Fig. 4, Supplement S6: Tables S6 and S10). When validated within population, stratified samples based on MAT performed best, diverging by −6.9% to −2.3% ($CI_{99}$) from the RMSE according to the reference random sampling from the entire population (Fig. 4, Supplement S6: Tables S7 and S11). The alternative NA treatment had little effect on these results in general but led to an influence of 49% of sampling procedures with species-specific calibration validated within sample, while systematically balanced samples performed best with site-specific calibration (Supplement S6: Fig. S4, Supplement S6: Tables S12–S15). Note that for the site-specific calibration, these sampling procedures refer to the allocation of observations for the 5-fold cross-evaluation, whereas for the species-specific calibration they refer to the selection of sites.

Sample size effects on model performance were very small but showed that more observations per free model parameter led to smaller RMSE, except with site-specific calibration when NA-producing runs were excluded (Fig. 4, Supplement S6: Fig. S5, Supplement S6: Tables S5–S8 and S12–S15). Among the size proxies relative (average) number of observation (N:q or $\bar{N}$:q) and site ratio (s:S), only s:S with species-specific calibration validated within population explained

more than 0.15% of the variance in RMSE, namely 1.0% (Fig. 4, Supplement S6: Table S8). An increase of 10% in N:q reduced the RMSE by approximately −0.5% to −0.1% with site-specific calibration, while an increase of 10% in the $\overline{\text{N}}$:q led to reductions of approximately −1.8% to −1.2% or −1.0% to −0.4% ($CI_{99}$) with respective species-specific calibration validated within sample or population (Supplement S6: Tables S5–S7). A 10% increase in s:S with species-specific calibration increased the RMSE by +0.08% to +0.13% ($CI_{99}$) if validated within sample and decreased it by −0.30% to −0.26% ($CI_{99}$) if validated within population (Supplement S6: Tables S6 and S7). By excluding NA-producing and non-converging runs, a 10 % increase in N:q increased the RMSE in site-specific calibration by +0.4 to +0.6% ($CI_{99}$; Supplement S6: Fig. S4, Supplement S6: Tables S12–S15).

Sites and species were included as grouping variables for the random effects and had little influence on model performance, except for sites with site-specific calibration. Sites explained 8.4% of the variance in RMSE with site-specific calibration and hence had a larger influence than sampling procedure and sample size (i.e. N:q) combined (Fig. 4, Supplement S6: Table S8). Species only explained more than 0.1% of the variance in species-specific calibration validated within sites, namely 0.4%. Thus, species had a slightly greater influence than $\overline{\text{N}}$:q and s:S combined (Fig. 4, Supplement S6: Table S8). When only converged calibration runs without NAs were analyzed, sites became the second most important driver of the variance in RMSE in site-specific calibration, explaining 27% (Supplement S6: Fig. S4, Supplement S6: Table S15).

## 3.2 Model projections

We analyzed the effects on the 100-year shifts at site level ($\Delta_{100}$) in autumn phenology projected with site- and species-specific models within sample or population and based on five random samples per projection mode. Each sample consisted of $10^5$ projected shifts per climate projection scenario, i.e. $2.6 \times 10^6$ projected shifts per sample, which were drawn from corresponding datasets that consisted of $>4.1 \times 10^7$ and $>1.9 \times 10^8$ or $>1.5 \times 10^9$ $\Delta_{100}$ projected with site- and species-specific phenology models for the sites within sample or population, respectively. These datasets were based on 16 climate model chains (CMC) based on the representative concentration pathway 4.5 (RCP 4.5), and 10 CMC based on RCP 8.5. The analyzed data (i.e. $3.9 \times 10^7$ $\Delta_{100}$) provided significant and decisive evidence against H0 for most estimated effects and most influences (Supplement S6: Tables S22–S25).

### 3.2.1 Estimated effects

Climate projection scenarios had the largest and second largest influence on projected autumn phenology in general, and the warmer RCP 8.5 caused larger shifts than the cooler RCP 4.5. Climate projection scenarios (i.e. unique combinations of RCP and CMC) explained between 46% and 64% of the variance in $\Delta_{100}$ in all projections (Fig. 5, Supplement S6: Table S22). The $\Delta_{100}$ according to site-specific models was +10.0 to +10.3 days ($CI_{99}$) based on the reference RCP 4.5 & CMC 1. This base $\Delta_{100}$ was altered between −11.1 to −11.0 days and ~−1.4 days ($CI_{99}$) by the RCP 4.5 scenarios, whereas the RCP 8.5 scenarios changed it by between −2.2 to −2.1 and +7.3 to +7.4 days ($CI_{99}$; except for the RCP 4.5 and CMC 2, which altered base $\Delta_{100}$

by +34.7 to +35.0 days, $CI_{99}$; Fig. 5, Supplement S6: Tables S19 and S23). The $\Delta_{100}$ according to species-specific models ranged from +11.4 to +11.6 days ($CI_{99}$) or from +8.2 to +8.4 days ($CI_{99}$) based on the reference RCP 4.5 and CMC 1 in respective projections within sample or within population (Fig. 5, Supplement S6: Tables S20, S21, S24, and S25). These base $\Delta_{100}$ were altered between −8.8 to −8.7 days and ±0.0 days ($CI_{99}$) or between −9.4 to −9.2 days and ±0.0 days ($CI_{99}$) by the RCP 4.5 scenarios, in corresponding projections within sample or within population (except for RCP 4.5 and CMC 2, which altered base $\Delta_{100}$ by >+37.1 days). The RCP 8.5 scenarios changed the base $\Delta_{100}$ projected within sample or within population by between −1.8 to −1.7 days and +5.8 to +6.0 days ($CI_{99}$) or between −2.0 to −1.8 days and +6.1 to +6.2 days ($CI_{99}$), respectively.

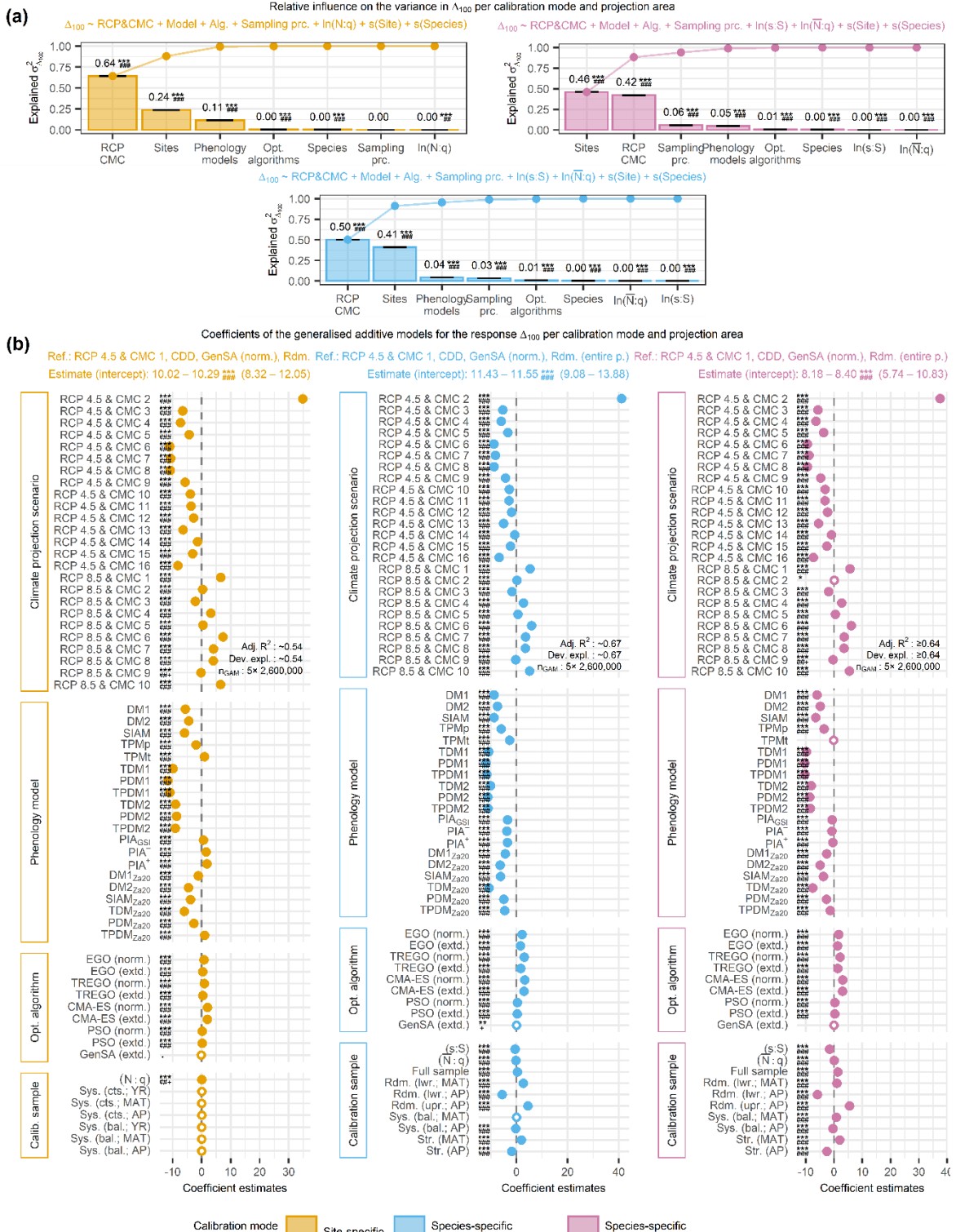

**(a)** Relative influence on the variance in $\Delta_{100}$ per calibration mode and projection area

$\Delta_{100}$ ~ RCP&CMC + Model + Alg. + Sampling prc. + ln(N:q) + s(Site) + s(Species)

$\Delta_{100}$ ~ RCP&CMC + Model + Alg. + Sampling prc. + ln(s:S) + ln($\overline{N}$:q) + s(Site) + s(Species)

$\Delta_{100}$ ~ RCP&CMC + Model + Alg. + Sampling prc. + ln(s:S) + ln($\overline{N}$:q) + s(Site) + s(Species)

**(b)** Coefficients of the generalised additive models for the response $\Delta_{100}$ per calibration mode and projection area

Ref.: RCP 4.5 & CMC 1, CDD, GenSA (norm.), Rdm.
Estimate (intercept): 10.02 – 10.29 ### (8.32 – 12.05)

Ref.: RCP 4.5 & CMC 1, CDD, GenSA (norm.), Rdm. (entire p.)
Estimate (intercept): 11.43 – 11.55 ### (9.08 – 13.88)

Ref.: RCP 4.5 & CMC 1, CDD, GenSA (norm.), Rdm. (entire p.)
Estimate (intercept): 8.18 – 8.40 ### (5.74 – 10.83)

Calibration mode (projection area):
- Site-specific
- Species-specific (projection within sample)
- Species-specific (projection within population)

**Figure 5: The relative variance in the projected 100-year shifts at site level ($\Delta_{100}$) of autumn phenology and corresponding effects explained by climate projection scenarios (i.e. representative concentration pathways, RCP, and climate model chains, CMC), phenology models, optimization algorithms, and calibration samples (i.e. sampling procedures and sample sizes).** The relative variance was estimated from analyses of variance (a) based on generalized additive models (GAM; b) for $\Delta_{100}$ according to site-specific models (left) as well as according to species-specific models when projected within sample (middle) or within population (right). In (a), the black error-bars indicate the range of estimated influence from the five GAM based on random samples. In (b), the median coefficient estimates from the five GAMs are visualized. If all five estimates were significant ($p < 0.005$), the median is indicated with a dot and with a circle otherwise. None of the 99% confidence intervals from any of the five GAMs extended beyond either dot or circle and are thus not shown. Note that coefficients estimate the difference to the reference $\Delta_{100}$. Thus, a negative coefficient estimate may indicate a projected advance or delay in autumn phenology, depending on how it relates to the reference. The abbreviations for the climate projection scenarios, phenology models, optimization algorithms, and sampling procedures in all figures are explained in respective Tables Supplement S1: Table S5, Supplement S2: Table S1, Supplement S4: Table S1, and Supplement S4: Table S2/S3. For a further description, see Fig. 4a and 4b.

When analyzed per phenology model, climate projection scenarios exhibited the largest influence on autumn phenology projected by over one-third of the models. Across the site-specific models, climate projections were most influential on $\Delta_{100}$ for 13 models and the largest fractions were observed for the models DM2 (76%) and DM2$_{Za20}$ (69%; Supplement S6: Fig. S6; Supplement S6: Table S27). Across the species-specific models, climate projections within sample or population were most influential on $\Delta_{100}$ for 12 or 8 models, respectively, and the largest fractions were observed for the models DM2 (49% or 58%) and DM2$_{Za20}$ (49% or 56%; Supplement S6: Fig. S7 and S8; Supplement S6: Tables S28 and S29).

Phenology models had the third and fourth largest influence on projections of autumn phenology according to site- and species-specific models, respectively, while TDM, PDM, and TPDM models generally projected the most pronounced forward shifts and CDD, TPMt and PIA models the most pronounced backward shifts. Phenology models explained 11% and 4% or 5% of the variance in $\Delta_{100}$ when projected with site- and species-specific models within sample or population, respectively (Fig. 5, Supplement S6: Table S22). When projected by site-specific models, the $\Delta_{100}$ based on the reference CDD model was reduced the most with the PDM1, TPDM1, and TDM1 models (from −11.8 to −9.8 days; CI$_{99}$; Fig. 5, Supplement S6: Tables S19 and S23). The largest increases occurred with the PIA$^+$, PIA$^-$, TPDM$_{Za20}$ models (from +0.9 to +1.8 days, CI$_{99}$). When projected with species-specific models, the $\Delta_{100}$ based on the reference CDD model was reduced with all other models (Fig. 5, Supplement S6: Tables S20, S21, S24, and S25). Here, the largest reductions occurred with the PDM1, TPDM1, and PDM2 or TDM1 models (from v12.0 to −11.2 or from −10.4 to −9.7 days; CI$_{99}$), while the smallest reductions occurred with the PIA$^-$, PIA$^+$, PIA$_{GSI}$, and TPMt models (from −3.8 to −2.6 days or from −0.8 to −0.1 days, if projected within sample or population respectively; CI$_{99}$).

Optimization algorithms had little influence on projections in general, while the algorithms CMA-ES (norm. and extd.) and TREGO (norm.) led to the largest deviations from the reference. Optimization algorithms explained less than 1% of the variance in $\Delta_{100}$ according to either site- or species-specific models (Fig. 5, Supplement S6: Table S22). When projected with site-specific models, the $\Delta_{100}$ according to the reference GenSA (norm.) was only reduced by the GenSA (extd.; ∼−0.1 days; CI$_{99}$) algorithm and increased most, namely between +0.9 and +2.1 days (CI$_{99}$), by both CMA-ES and the TREGO (norm.) algorithms (Fig. 5, Supplement S6: Tables S19 and S23). When based on species-specific models, the lowest $\Delta_{100}$ was obtained with the reference. Again, both CMA-ES and the TREGO (norm.) algorithms increased $\Delta_{100}$ the most compared to the

reference, namely from +3.0 to +3.3 days or +2.1 to +3.1 days ($CI_{99}$) in projections within sample or within population (Fig. 5, Supplement S6: Tables S20, S21, S24, and S25).

Sampling procedures had by definition no influence on projections with site-specific models and the third or fourth largest influence on projections with species-specific models. Since site-specific model parameters for projections were calibrated with all observations per site, effects of corresponding sampling procedures on $\Delta_{100}$ would be random. Subsequently, our results indicated no general (i.e. according to *all* five samples) significant or decisive effect of any sampling procedure (Fig. 5, Supplement S6: Table S23). However, the *p*-values of two sampling procedures fell below the significance level according to at least one of the five GAMs, leading to a type I error or "false positive", whereas none of the GAMs resulted in a decisive influence according to the Bayes factor. In projections with species-specific models, sampling procedures had an influence and explained 3% or 6% of the variance in $\Delta_{100}$ when projected within site or within population (Fig. 5, Supplement S6: Table S22). In comparison to the reference random sample from the entire population, $\Delta_{100}$ was reduced or increased the most when projections were based on random samples from the lower or upper half of the population according to average autumn phenology, respectively (Fig. 5, Supplement S6: Tables S20, S21, S24, and S25). Corresponding effect sizes were ~−5.6 days or ~+4.5 days and −5.9 to −5.8 days or +5.4 to +5.5 days ($CI_{99}$)) when projected within sample and within population, respectively.

Sample size proxies had the smallest influence. Neither N:q and $\overline{N}$:q nor the site ratio (s:S) explained more than 0.5% of the variance in $\Delta_{100}$ (Fig. 5, Supplement S6: Table S22). In projections with site-specific models, effects were ~+0.0 days for N:q ($CI_{99}$; Fig. 5, Supplement S6: Tables S19 and S23). In projections with species-specific models, the effects were ~+0.2 days or ~+0.1 days for $\overline{N}$:q ($CI_{99}$) when projected within sample and within population, respectively, and +0.0 days for s:S ($CI_{99}$; Fig. 5, Supplement S6: Tables S20, S21, S24, and S25).

Sites were the most and second most important driver of autumn phenology projected with both site- and species-specific models, while the influence of species was very low. Sites explained 24% and 41% or 46% of the variance in $\Delta_{100}$ when projected with site- and species-specific models within samples or population, respectively (Fig. 5, Supplement S6: Table S22). Species accounted for less than 0.5% of the variance in $\Delta_{100}$ projected with either site- or species-specific models.

### 3.2.2    Relations with model performance

Coefficient estimates for performance and projections were negatively correlated across phenology models and positively correlated across optimization algorithms for both site- and species-specific models, but with neither decisive nor significant evidence. With site-specific calibration and projection, we derived the highest Kendall rank correlations for sampling procedures with $\tau$ = +0.71 (*p* = 0.024 and $\underline{BF}_{01}$ = 1/5.1) and weaker negative and positive correlations for phenology models and optimization algorithms, respectively (Fig. 6, Supplement S6: Table S26). With species-specific calibrations and projections, the correlations for phenology models and sampling procedures were negative, whereas those for optimization algorithms were positive. When projected within sample or population, the strongest correlations were derived for optimization algorithms or for phenology models, namely $\tau$ = −0.50 or $\tau$ = −0.35 (*p* = 0.061 and $\underline{BF}_{01}$ = 1/2.2 or *p* = 0.032 and $\underline{BF}_{01}$ = 1/3.1),

respectively (Fig. 6, Supplement S6: Table S26). Thus, while the best performing phenology models were related to larger $\Delta_{100}$, the best performing optimization algorithms were associated with smaller $\Delta_{100}$, without any regularity in sampling procedures. In other words, autumn phenology was projected to occur later if based on better performing models.

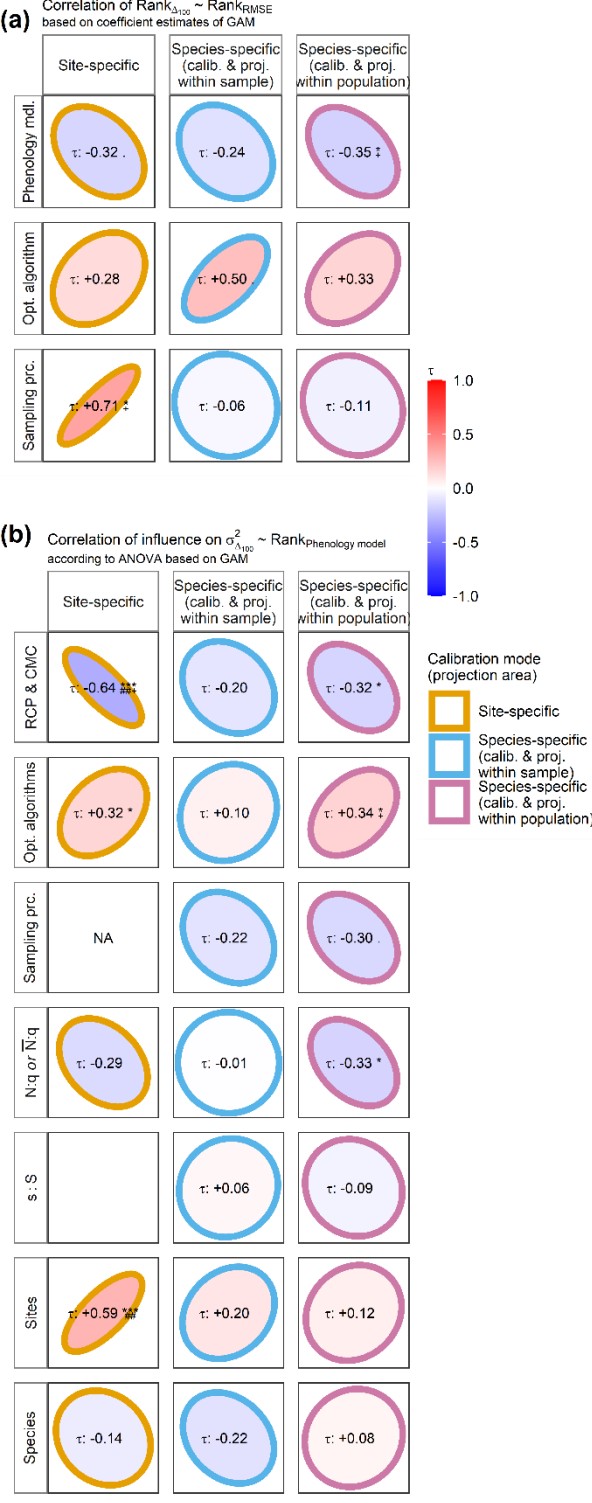

**(a)** Correlation of Rank$_{\Delta_{100}}$ ~ Rank$_{RMSE}$
based on coefficient estimates of GAM

|  | Site-specific | Species-specific (calib. & proj. within sample) | Species-specific (calib. & proj. within population) |
|---|---|---|---|
| Phenology mdl. | τ: -0.32 . | τ: -0.24 | τ: -0.35 ‡ |
| Opt. algorithm | τ: +0.28 | τ: +0.50 . | τ: +0.33 |
| Sampling prc. | τ: +0.71 ‡ | τ: -0.06 | τ: -0.11 |

τ
1.0
0.5
0.0
-0.5
-1.0

**(b)** Correlation of influence on $\sigma^2_{\Delta_{100}}$ ~ Rank$_{Phenology\ model}$
according to ANOVA based on GAM

|  | Site-specific | Species-specific (calib. & proj. within sample) | Species-specific (calib. & proj. within population) |
|---|---|---|---|
| RCP & CMC | τ: -0.64 ‡‡‡ | τ: -0.20 | τ: -0.32 * |
| Opt. algorithms | τ: +0.32 * | τ: +0.10 | τ: +0.34 ‡ |
| Sampling prc. | NA | τ: -0.22 | τ: -0.30 . |
| N:q or N̄:q | τ: -0.29 | τ: -0.01 | τ: -0.33 * |
| s : S |  | τ: +0.06 | τ: -0.09 |
| Sites | τ: +0.59 ‡‡‡ | τ: +0.20 | τ: +0.12 |
| Species | τ: -0.14 | τ: -0.22 | τ: +0.08 |

Calibration mode
(projection area)

Site-specific

Species-specific
(calib. & proj.
within sample)

Species-specific
(calib. & proj.
within population)

**Figure 6: Kendall rank correlation (τ) between coefficient estimates of explanatory variables for log-transformed root mean square error (RMSE) and 100-year shifts ($\Delta_{100}$) at site level (a) and between coefficient estimates of phenology models for log(RMSE) and the relative influence on the variance in $\Delta_{100}$ of the remaining factors (b).** The color of the ellipses corresponds to the calibration mode and corresponding projections, whereas their filling visualises the value of τ. The value of τ is further expressed by the angle (negtive vs. postive) and length of minor axis (absolute value). Asterisks and dots refer to the *p*-value, while hashtags and crosses refer to the minimum Bayes factor (see Fig. 4 for further details).

Projections with site-specific models were influenced more by climate projection scenarios and less by sites when based on better performing phenology models. The evidence was strongest for the correlations between the performance rank of phenology models and the relative influence of climate projection scenarios on the variance in $\Delta_{100}$ ($\tau = -0.64$; $p < 0.005$ and $\underline{BF}_{01} = 1/788$) or sites ($\tau = +0.59$; $p < 0.005$ and $\underline{BF}_{01} = 1/249$; Fig. 6, Supplement S6: Fig. S5 and S6, Supplement S6: Table S26). This suggests that the better the underlying models performed, the more closely the projections of autumn phenology followed the climate signal, and vice versa. Also, the better the underlying models performed, the more the projections of autumn phenology became detached from the sites.

Projections with species-specific models and within population were a little more influenced by climate projection scenarios and less influenced by optimization algorithms when based on better performing models, whereas the influence of factors on projections within sample appeared to be unrelated to model performance. When projected within population, the evidence was strongest for the correlations between performance rank of models and climate projection scenarios ($\tau = -0.32$; $p = 0.040$ and $\underline{BF}_{01} = 1/2.6$) or optimization algorithms ($\tau = +0.34$; $p = 0.030$ and $\underline{BF}_{01} = 1/3.3$; Fig. 6, Supplement S6: Fig. S5, S7, and S8, Supplement S6: Table S26). That is, the better the models are, the more the projections depended on the climate signal and the less they were influenced by the optimization algorithm.

## 4    Discussion

We evaluated the effects of phenology models, optimization algorithms, sampling procedures, and sample size on the performance of phenology models calibrated per site (i.e. one set of parameters per species and per site) and per species (i.e. one set of parameters per species and for the entire population). The performance was mainly influenced by the phenology models, followed by the optimization algorithms. In general, simple phenology models that depended on daily temperature, day length, and partly on average seasonal temperature or spring leaf phenology performed best, and non-Bayesian optimization algorithms outcompeted the Bayesian algorithms. The entire population was best represented by species-specific phenology models calibrated with stratified samples that were based on equally sized bins according to the average phenology per site. Site- or species-specific models performed best when trained with systematically balanced or stratified samples based on mean annual temperature, respectively. The bias-variance trade-off (i.e. overfitting) with site-specific calibration increased considerably when the ratio of the number of observations relative to the number of the free model parameter fell below nine.

We further evaluated the effects of phenology models, optimization algorithms, sampling procedures, sample size, and climate projection scenarios on the projected 100-year shift in autumn leaf phenology according to site- or species-specific

models. Projected autumn phenology generally shifted between −13 and +12 days or between −4 and +20 days according to the representative concentration pathway 4.5 or 8.5 (RCP 4.5 or RCP 8.5, respectively), depending on the scenario and phenology model and based on the reference optimization algorithm and sampling procedure. The shifts were mainly influenced by climate projection scenarios and sites. The relative influence of phenology models was surprisingly small, but shifts projected with better performing models were generally larger and depended more on projected climate change than shifts according to models with poorer performance.

## 4.1    Phenology models

Phenology models had the largest influence on model performance. In model comparisons, better performing models are usually accredited as being based on relevant and correctly interpreted processes (e.g., Delpierre et al., 2009; Keenan and Richardson, 2015; Lang et al., 2019; Zani et al., 2020). Here we show that phenology models exerted the largest influence on model performance of all the factors analyzed. This reinforces model comparisons to identify relevant processes of phenology.

Relatively simple models driven by daily temperature, day length, and partly by seasonal temperature or spring leaf phenology performed best. Patterns in ranked coefficient estimates generally showed that the models DM1 and DM2 developed by Delpierre et al. (2009) and the models $DM1_{Za20}$, $DM2_{Za20}$, $TDM_{Za20}$, and $SIAM_{Za20}$ adapted by Zani et al. (2020) performed best. These models are very similar to the models that Liu et al. (2020) adapted from Delpierre et al. (2009) and Caffarra et al. (2011), which performed best in their model comparison study. Further, the models DM1 and DM2 performed best in Delpierre et al. (2009). However, the models $DM1_{Za20}$, $DM2_{Za20}$, $TDM_{Za20}$, and $SIAM_{Za20}$ did not lead to the best results in Zani et al. (2020). Daily senescence rate in all these models depends on daily temperature and day length, while the threshold is either a constant or linearly derived from the actual average temperature during the growing season or the site-anomaly of spring phenology. Interestingly, the best performing models in site-specific calibration were those adapted by Zani et al. (2020) such that the senescence rate was based on a sigmoid curve, which economized one free model parameter (Table 1 and Supplement S2: Table S1). We hypothesize that fewer parameters generally lead to an advantage with few training observations, which needs to be examined in more detail in further studies. Finally, our study supports previous studies that have also demonstrated the superiority of models based on daily temperature, day length, seasonal temperature, and spring phenology, while questioning the effect of photosynthesis on autumn leaf phenology as suggested by Zani et al. (2020) and indicating that the model by Lang et al. (2019) benefits from considering seasonal drivers.

Sites had a relatively large influence on projections with species-specific models, and the number of sites per sample had a negative effect on the sample-level model performance of species-specific models. Relevant drivers may be missed by models based on senescence rates driven by temperature and day length and a corresponding threshold (e.g. Fu et al., 2014b). Recent models accounted for this and based their threshold for the senescence rate on spring phenology (SIAM model; Keenan and Richardson, 2015) or on seasonal drivers such as the average growing season temperature or accumulated net photosynthetic product (TDM or PIA models; Liu et al., 2019; Zani et al., 2020). However, Gill et al. (2015) and Chen et al. (2018) observed site-specific responses of leaf phenology to climate change, which could be due to site-specific soil properties

(Arend et al., 2016), nutrient availability (Fu et al., 2019), and local adaptation (Peaucelle et al., 2019), which are not yet included in the current models. In addition, observations may be biased (Liu et al., 2021) and the perceptions of observers at different sites are usually not aligned. Models can consider relevant but excluded drivers and observer bias through differently calibrated parameters, such as a sample-specific threshold constants, for example. However, the positive effect of such sample-specific parameters decreases as the number of sites in the sample increases, shifting the modeled values closer to the mean observation of the calibration sample. Consequently, we expect (1) a larger effect of sites on projections based on species-specific rather than site-specific models and (2) an increasing RMSE with more sites per sample. Here we observed both, i.e. the relative influence of sites on projections increased from 24% to 46% or 41% if based on site-specific models or species-specific models projected within sample or within the entire population, respectively. Moreover, the RMSE of species-specific models validated within sample increased with more sites as expressed by the site ratio s:S. This demonstrates that some relevant drivers of autumn leaf phenology are not yet considered in the evaluated process-oriented models and/or that bias in observed phenology data may be amplified when the same observer is at a particular site for multiple years.

## 4.2    Optimization algorithms

We generally obtained the best results with the GenSA algorithm and found that the Bayesian algorithms ran in extensive mode outperformed those ran in normal mode. While Simulated Annealing and its derivatives have been the algorithms of choice in various studies that calibrated process-oriented models of tree leaf phenology (e.g., Chuine et al., 1998; Meier et al., 2018; Zani et al., 2020), one derivative, namely GenSA (Xiang et al., 2013), is further the default algorithm of the R package phenoR (Hufkens et al., 2018). In this study, GenSA generally delivered the best results, which confirmed this choice. However, our results depended on the control settings of the algorithms such as the number of iterations. The Bayesian algorithms EGO and TREGO always performed better when executed in extensive mode and may lead to better results if the iterations and/or the number of starting points were increased further.

However, more iterations did not lead to more accurate results for all optimization algorithms. We basically applied off-the-shelf algorithms (Trautmann et al., 2011; Bendtsen, 2012; Xiang et al., 2013; Picheny and Ginsbourger, 2014; Dupuy et al., 2015; Hufkens et al., 2018), using the respective default configurations except for the control of the number of iterations, which we adjusted depending on the number of free model parameters to execute them in normal and extended mode (i.e. few and many iterations). Expecting that more iterations lead to better results, we were surprised to find that this was not always true. However, all studied algorithms sample solutions of the cost function by changing the free parameters in small steps. This step size depends, among others, on the search space and the number of iterations. In turn, the complexity of the cost function depends on the model and the number of free parameters. While many iterations lead to small steps and vice versa, small steps may cause the algorithm to get stuck in a local optimum whereas large steps may cause it to overstep the global optimum (i.e. the exploration-exploitation trade-off; Maes et al., 2013; Candelieri, 2021). In addition, larger samples and more free parameters are expected to lead to more local optima. Therefore, we strongly suggest that studies of process-based

phenology models carefully set and test the control parameters of the optimization algorithms in dependence of the models as well as communicate and discuss the applied settings.

## 4.3    Calibration samples

Stratified selected sites based on autumn phenology seemed to represent the population better than the population itself, which appears to be an artefact of a model bias towards the mean phenology. We evaluated the degree to which species-specific phenology models represented the entire population according to the ratio of the external *sample* RMSE to the external *population* RMSE. This ratio lay above one for stratified samples based on autumn phenology. In other words, the external RMSE decreased when calculated for the entire 500 sites instead of the 17 sites. This finding can be explained by the fact that modelled values tend towards the mean predictand (e.g., visible in Delpierre et al., 2009, Fig. 2, and in Lang et al. 2019, Fig. 4). Given such a tendency, the errors between modelled and observed values result in a U-shaped (i.e. convex) curve across the predictands (i.e. smaller errors for the mean predictand and larger errors for the extremes). Consequently, normally distributed predictands result in a smaller RMSE than for example uniformly distributed predictands because the former distribution accounts for relatively more small errors around the mean predictand than the latter. Here we argue that autumn phenology (i.e. the predictand) tends to follow a uniform distribution in stratified samples based on autumn phenology and normally distributed in the full sample. Further, the tendency of phenology models towards the mean observed phenology manifested in smaller variances of modelled than observed phenology in Delpierre et al. (2009, Fig. 2), Keenan and Richardson (2015, Fig. 3), Lang et al. (2019, Fig. 4), Liu et al. (2019, Fig. 2), and Zani et al. (2020; Fig. 3D, only for some models). Thus, we suggest that our phenology models were biased towards the mean, too, which led to the seemingly better representation of the population by stratified samples based on autumn phenology. Moreover, we hypothesize that the RMSE ratio in the other samples did not exceed one because the distributions of autumn phenology tended towards normal in all these samples. Finally, it follows from the above line of thought that models with a tendency towards the mean predictand have an advantage over models without such a tendency when the RMSE is calculated from normally distributed observations (provided convex and uniform error curves cover the same area within the observation range). This seems to be a hindrance in the search for models that represent a population and thus should fit all observations equally. Therefore, we advocate the use of stratified samples based on autumn phenology to test and evaluate calibrated models at the population level.

Our results suggest the use of systematically balanced observations in cross-validation and of randomly selected sites from stratified bins based on mean annual temperature in species-specific modelling. Systematically continuous samples led to the best results of cross-validated models, if NA values and non-converging runs were penalized and included, whereas they were outperformed by systematically balanced samples based on autumn phenology or year, if NA-producing and non-converging runs were excluded. Thus, the exclusion of such runs benefitted the balanced samples and/or penalized the continuous samples. This may be the case when some balanced samples led to more NAs or non-converging calibration runs than the continuous samples and/or some continuous samples led to very small root mean square errors despite one or more NAs. Since the first possibility seems more likely, we suggest that the best cross-validations are obtained with systematically

balanced observations based on phenology or year, but this may also lead to convergence problems. In addition, we studied the effects of site selection on model calibration. Not surprisingly, if validated within sampled sites, the best results were obtained with samples of more uniform phenological patterns, namely with randomly sampled sites from either half of the population according to autumn phenology. More interestingly, if validated within the sites of the entire population, models calibrated with 12 randomly selected sites from stratified equally sized bins based on mean annual temperature outperformed models calibrated with all 500 sites (i.e. full sample) and led to lowest RMSE. This result is in line with the conclusions by Cochran (1946) and Bourdeau (1953) that random components are beneficial in ecological studies and stratified sampling represents a population equally or better than random sampling. Even more remarkable is that the calculation time of the calibrations for models trained with these stratified samples was notably shorter than for models trained with the full sample. Therefore, we recommend stratified sampling based on mean annual temperature for the calibration of species-specific models, since it leads to the best model performance at the population level while requiring very little computational resources.

Sample size effects suggested to calibrate with at least nine observations per free model parameter (N:q) to prevent serious overfitting. We estimated the degree of the bias-variance trade-off with the ratio between external and internal RMSE (Cawley and Talbot, 2010, Sect. 5.2.1). Our results showed that, while this ratio decreases constantly, the rate of decrease changes notably between N:q = ~9 to ~12. This range is in line with the sometimes mentioned rule of thumb of at least N:q = 10 in regression analysis. Besides and more specific, Jenkins and Quintana-Ascencio (2020) suggest the use of at least N:q = 8 in regression analysis if the variance of predictands is low and at least N:q = 25 when it is high. Although we calibrated process-oriented models and not regression models, these minimum N:q are in line with our study and thus seem to apply, too. Moreover, while N:q = 9 appear to be the minimum, we suggest the use of larger N:q if the model is calibrated with observations derived from more than one site to account for an increased variance.

The RMSE increased for some models when non-converging calibration runs or runs that yielded NA values in either the calibration or validation were excluded. For our performance results, we substituted NA values with an error between observed and simulated day of autumn leaf phenology of 170 d (i.e. a larger error than observed in any calibration run). Accordingly, non-converging runs led to an RMSE of 170 d, which were analyzed together with the RMSE of converging runs (cf. Sect. 2.5.2). Now, if the RMSE is analyzed excluding the non-converging runs and the runs that yielded NA values, intuitively one would expect the average RMSE to shrink, but this was not the case in the species-specific models (cf. Supplement 6: Sect. S2.2.2). In other words, punishing with large RMSE and large errors for non-converging runs and NA values led to smaller estimated RMSE and thus better estimated model performance. The relationship between performance and sample size may explain this counterintuitive result. Large samples favored the performance of species-specific models but also more often led to NA values than smaller samples (cf. Supplement 6: Sect. S1). At the same time, larger samples weaken the effect of a particular substitution of an NA value with 170 d on the RMSE. Thus, calibrations with large samples may well have been more accurate despite some NA values and may have resulted in lower RMSE despite NA substitution, which positively affected the overall performance of the models.

## 4.4 Projections of autumn phenology

Overall, the climate projection scenarios were the primary drivers of the projected shifts in autumn phenology, with the warmer scenario causing later autumn phenology than the cooler scenario, which is consistent with the currently observed main effect of climate warming. Having the largest influence in two out of three projection modes, climate projection scenarios explained between 46% and 64% of the variance in the 100-year shifts of autumn phenology. On average, the projected autumn phenology occurred 8–9 days later when projected with the warmer RCP 8.5 than with the cooler RCP 4.5 scenarios, which corresponds to the observed main effect of warming. Past climate warming was found to mainly delay autumn phenology (Ibáñez et al., 2010; Meier et al., 2021), but slight forward shifts or a distribution around a temporal change rate of zero have also been observed (Menzel et al., 2020; Piao et al., 2019). Such inconsistent past trends may be explained by the fact that autumn phenology (i.e. observed with canopy greenness rather than chlorophyl content; cf. Sect. 2.1.1 and Mariën et al., 2021) depends more on the severity than the type of weather event, with, for example, moderate heat spells causing backward shifts but extreme heat spells and drought causing forward shifts (Xie et al., 2015). Since the number and severity of heat spells is related to sites (e.g. warmer lowland vs. cooler highland sites; Bigler and Vitasse, 2021), such opposing effects of weather events may explain the large influence of sites on projected shifts in autumn phenology, as discussed below. In addition, the length of the growing season is affected by shifts in spring and autumn phenology for deciduous trees. Our projections were based on spring phenology that advanced by 20 days within 100 years. Subsequently, the projected growing season lengthened by 7–32 days (RCP 4.5) or by 16–40 days (RCP 8.5), even when autumn phenology shifted forward, as projected with some models and discussed further below. Therefore, our study supports a general lengthening of the growing season due to projected climate warming, as also suggested by Delpierre et al. (2009), Keenan and Richardson (2015), and Meier et al. (2021), in contrast to Zani et al. (2020).

The divergent autumn phenology projections of the scenario RCP 4.5 and CMC 2 might be due to temperature and precipitation biases in this scenario. We based our autumn phenology projections on 26 different climate model chains of the EURO CORDEX dataset, each consisting of a global and regional climate model (GCM and RCM, respectively; Supplement S1: Table S5). While the patterns in the projections generally matched our expectations, the projected autumn phenology based on the scenario RCP 4.5 and CMC 2 behaved differently from those based on the other scenarios (Fig. 5). The EURO CORDEX is a state-of-the-art GCM and RCM ensemble and widely accepted to contain high quality climate projection data despite some biases (Coppola et al., 2021; Vautard et al., 2021). The afore mentioned RCP 4.5 and CMC 2 stands for the RCM CNRM-ALADIN63 (ALADIN63; Nabat et al., 2020) driven by the GCM CNRM-CERFACS-CNRM-CM5 (CM5; Voldoire et al., 2013) under RCP 4.5 (version two of the run r1i1p1; Supplement S1: Table S5). This scenario is the only one based on the RCM ALADIN63 in this study, whereas five other scenarios were also based on the GCM CM5. ALADIN63 is the successor of CNRM-ALADIN53 (ALADIN53; Colin et al., 2010), which emerged after ten years of development and is arguably different from the latter but still related to it (Nabat et al., 2020). The GCM-RCM chain CM5 - ALADIN53 had comparatively great difficulty in accurately modeling the interannual and seasonal variability of mean temperature, seasonal precipitation for

summer (i.e., June, July, and August; JJA), and the seasonal variability of different drought indices in a comparison of the historical runs from the EURO-CORDEX ensemble with observational data for southern Italy (Peres et al., 2020). Another comparison resulted in relatively large errors in the minimum temperature during summer (JJA) for both ALADIN53 and ALADIN63 driven by CM5 as well as in the second largest deviation from the observed temperature trend for CM5 - ALADIN63 (i.e. more than twice the temperature increase during 1970–2005 as observed) based on historical runs of the EURO-CORDEX ensemble and weather observations for the Pannonian Basin in the southeast of Central Europe (Lazić et al., 2021). Since we treated all GCM-RCM used in this study the same (Supplement S1: Sect. 2), it may be that such biases in temperature variability and summer temperature are responsible for the deviating projections of autumn phenology when based on RCP 4.5 and CMC 2.

Sites generally exhibited the second largest influence on projected shifts and had more influence when the latter were projected with species- than with site-specific models, which could be due to correct modelling of site differences or to poorer calibration for sites with phenology that deviates from the sample mean. Studies of past changes in autumn phenology of trees found ambiguous trends between different sites (Piao et al., 2019; Meier et al., 2021). Different trends may be the result of opposing weather effects, e.g., moderate versus severe heat and drought spells (Xie et al., 2015), or of different correlations between spring and autumn phenology, dependent on nutrient availability and elevation (Fu et al., 2019; Charlet de Sauvage et al., 2022), possibly related to local adaptation (Alberto et al., 2011; Quan and Wang, 2018). Thus, it may be that such opposing weather effects or different correlations led to the strong influence of sites on projected autumn phenology. However, this strong influence could also be due to the above-described tendency of models to the mean and subsequent poorer calibration for sites at the extremes of the climatic and phenological spectrum within samples. This hypothesis is further supported by the larger relative influence of sites on projections with species- than with site-specific models. In addition, species-specific models performed worse than site-specific models (based on converged runs without NA values), as was also reported by Basler (2016) with respect to models of spring phenology, while Liu et al. (2020) found that species-specific models poorly reflected the spatial variability of autumn phenology. Thus, it seems improbable that the species-specific models with generally poorer performance predicted site differences more accurately than the site-specific models with generally better performance. Therefore, we suspect that the large influence of sites on projections with species-specific models is primarily due to insufficiently modelled processes of leaf phenology and the consequent tendency of phenology models to the mean.

The influence of phenology models on projected autumn phenology was relatively low and the range of projections relatively small. The largest influence of phenology models was 11% and occurred in projections based on site-specific models and hence was almost six times smaller than the influence of climate projection scenarios. While the underlying processes differ between each model (Delpierre et al., 2009; Keenan and Richardson, 2015; Lang et al., 2019; Liu et al., 2019; Zani et al., 2020), the influence of these differences on the projected autumn phenology did not affect the projected lengthening of the growing season: Different models altered the reference shifts of +8.2 to +11.6 days by -12 to +2 days, which resulted in some forward shifts in autumn phenology with the cooler RCP 4.5 scenarios. Moreover, the difference between the models lay within

14 days (i.e. −12 to +2 days), which is less than the uncertainty attached to recordings of autumn phenology based on human observations (i.e. due to small sample sizes and observer bias; Liu et al., 2021). In other words, the different process-oriented models led to differences in the length of the growing season that were smaller than the uncertainty in the data upon which we based our projections. Therefore, our results justify the assumption, that the examined phenology models do not differ fundamentally in their underlying processes, even if we acknowledge that the TDM, PDM, and TPDM models (Liu et al., 2019) behaved differently than the other models (i.e. they resulted in the largest forward or smallest backward shifts of autumn phenology). Rather, we suggest that the effects of temperature and day length, which all analyzed models simplify in different ways, mostly suppress the effects of other concerned drivers.

Better-performing models generally projected later autumn leaf phenology, which may add a new dimension to the discussion about realistic projections of autumn leaf phenology, but should be treated with caution, as corresponding correlations were relatively low. The PIA models in Zani et al. (2020) projected autumn phenology to advance between the years 2000 and 2100 as a result of increased photosynthetic activity. This result has thus been debated (Norby, 2021; Zani et al., 2021; Lu and Keenan, 2022; Marqués et al., 2023) and could not be reproduced in our study. Here, we found positive and negative coefficient estimates, depending on the projection mode, which seems to be partly in line with Zani et al. (2020), but that is actually not the case: First, our estimates refer to changes relative to the reference, which was always positive. Thus, while the negative coefficients for the PIA models indicated negative deviations from the reference, they still resulted in later autumn phenology. Second, Zani et al. (2020) reported an advancement of autumn phenology only for projections based on PIA models and in contrast to the other models, while our results suggest the largest forward shifts for the TDM, PDM, and TPDM models (Liu et al., 2019). Moreover, the negative correlations between phenology models ranked by their performance and projection estimates showed that models with smaller root mean square errors generally projected larger shifts than models with larger root mean square errors. In other words, while we were unable to replicate the projected pronounced forward shifts in autumn phenology due to increased photosynthetic activity, our results suggest that better performing models tend to project a later autumn phenology.

## 4.5    Methodological issues

### 4.5.1    Driver data

Modelled weather data can be biased, which affects model outputs based on these data. For example, correcting climate projections for bias increased the accuracies of projected forest ecosystem function and of the simulated timing of leaf phenology (Drepper et al., 2022; Jourdan et al., 2021). Here, we refrained from bias-correcting the meteorological data for the past and future, which likely negatively affected the accuracy of the simulated timing of autumn leaf phenology for the past and future. Thus, we probably received too large RMSE and projected shifts that were both too small and too large. But did the use of uncorrected meteorological data affect our comparison of model vs. calibration effects on model performance and projections? The used meteorological data for the past is likely more accurate for some sites than for others. This is probably

also true for the used meteorological data for the future, but the sites with more vs. less accurate data likely differ between climate scenarios. In addition, some scenarios can be systematically warmer than others, for example (cf. above). Therefore, the effect of sites on model performance and the effect of climate scenarios on model projections was probably inflated by the uncorrected meteorological data. In contrast, these data probably affected all models similarly and the sampling procedures randomly, whereas the optimization algorithms remained unaffected. Thus, the use of uncorrected meteorological data most likely had little impact on our results.

Spatial and elevational differences between a particular site and the centre of the corresponding grid cell, from which the meteorological data were extracted, affect the input data. Gridded data may poorly represent the conditions at a particular site due to spatial and elevational differences. For example, precipitation and temperature can change in response to different terrain and the lapse rate, respectively, while the leaf area index and plant-available water capacity can change due to different vegetation and soil conditions. These effects of spatial and elevational differences were not considered in this study and may have led to inaccurate input data (e.g. average MAT for the site Grossarl, 47.2° N / 13.2° E at 900 m a.s.l., in the Austrian Alps was ~0.6° C, which makes beech growth unlikely; Holtmeier and Broll, 2020). The degree of inaccuracy probably differs between sites, which inflated the site effects on model performance and model projections. In contrast, the effects of models, sampling procedures, and optimization algorithms were probably unaffected by the inaccurate input data (cf. above), so these data most likely had a neglectable effect on our results.

### 4.5.2 Daily minimum vs. mean temperature as driver of the senescence rate

Most original publications of the compared models calculated the senescence rate from the mean rather than the minimum temperature, whereas we used the minimum temperature for all models. Our choice was based on the stress exhibited by cool temperatures that promotes autumn leaf phenology (Jibran et al., 2013; Lim et al., 2007) and the recent model comparison by Zani et al. (2020), who used daily minimum temperature throughout their study. This choice allowed to compare or study with Zani et al. (2020) and to assess the response curves of the senescence rate. However, inferences on the drivers of the senescence rate would be more profound, if they were based on a comparison that additionally considers models driven by mean temperature, as suggested in some of the original publications (i.e., Delpierre et al., 2009; Dufrêne et al., 2005; Keenan and Richardson, 2015; Liu et al., 2019). Such an extended comparison is certainly essential to gain further insight in the process of leaf senescence but may only focus on the models to remain feasible, rather than also including optimization algorithms and sampling procedures.

### 4.5.3 Treatment of NA values and non-converging calibration runs

Especially the ranks of site-specific phenology models and corresponding optimization algorithms were affected by the treatment of NA values and non-converging calibration runs. Here, we performed two analyses in which we either replaced NA values and non-converging runs with a fixed value or we excluded the affected runs. We are not aware of any other study involving process-oriented models of tree leaf phenology that has mentioned NA-producing or non-converging calibration

runs and their treatment. We doubt that our study is an exception in that it produced NAs or non-converging runs at all. Be that as it may, in the absence of previous studies addressing this issue, we could not refer to any established treatment and had to find a way to deal with NA values and non-converging runs. We chose (1) to penalize NAs with a replacement slightly larger than the largest modelled differences or (2) to exclude concerned runs. Since only "bad" runs were excluded, exclusion may bias the analysis by leading to overly optimistic results and replacement seemed preferable, especially for ranking factors by their effect on performance. However, replacement adds an artificial effect and thus affects the estimated effect sizes, why they should be compared with corresponding effect sizes after exclusion and treated with caution. The models with the most parameters also led to the most NA-producing or non-converging runs, especially with site-specific calibration performed with GenSA or Bayesian optimization algorithms. Subsequently, we found considerable differences between the ranking of phenology models and optimization algorithms depending on the treatments with replacement and exclusion.

### 4.5.4 Evaluation of model performance based on the root mean square error

We evaluated model performance solely based on the RMSE, which allows only limited conclusions to be drawn about biological processes but reveals the effects of various factors on model performance and projections. Small RMSE values may also result from poorly calibrated models, due to intercorrelation between free model parameters or limited data (e.g., regarding the base temperature and threshold in the CDD model or climatic and temporal sections of the habitat and longevity of a species, respectively; Chuine and Régnière, 2017). Therefore, Chuine and Régnière (2017) recommended to complement the quantitative evaluation with a qualitative assessment of the calibrated parameters and the resulting curves of the rate functions. Since we were less interested in the biological processes that drive leaf phenology and since we evaluated over 2 million different models, we refrained from a qualitative assessment and focused solely on the quantitative RMSE. Therefore, conclusions about the processes driving autumn phenology should be drawn from our study with caution. At the same time, the RMSE remains an important measure for model comparison and selection. Thus, our conclusions about the relationships between the factors analyzed and model performance or projections, and between model performance and projections, are relevant for future studies of leaf phenology.

### 4.5.5 Choice of generalized additive models instead of linear mixed-effects models

We opted for generalized additive models rather than linear mixed-effects models due to software limitations and thus benefited from higher computing speed without affecting the results. The amount of $\Delta_{100}$ data was too large and thus the variance-covariance matrix could not be solved by the LAPACK library (Anderson et al., 1999) integrated in R (R Core Team, 2022) and called upon by the function lme4::lmer (Bates et al., 2015). However, the data could be processed with the function mgcv::bam (Wood, 2011, 2017), which allowed an alternative formulation of a linear model with random intercepts. While mgcv::bam was much faster than lme4::lmer (personal observation), the corresponding coefficient estimates were practically identical (comparison not shown).

### 4.5.6 Evaluation based on sampled model projections

Since we feared hardware limitations if we evaluated all $\Delta_{100}$ data, we opted for an evaluation based on samples, the results of which are convincing. While we were able to fit the generalized additive models for all $\Delta_{100}$ data, the subsequent analysis of variance (functions aov and drop1 in the R package stats; R Core Team, 2022) required an enormous amount of computing power (>600 GB RAM per CPU-core for the smallest dataset, i.e. the projections based on site-specific calibration). Therefore, we reduced the amount of data to be processed by drawing samples, fitting a generalized additive model for each sample, and deriving an ANOVA from each model. Since the resulting coefficient estimates, confidence intervals, and estimates of the relative explained variance were within reasonable ranges, this procedure further strengthened our confidence in the results.

### 4.5.7 Significance level and Bayes factors

We chose a lower significance level than commonly used in ecology studies and complemented the *p*-value with the minimum Bayes factor to prevent over- or misinterpretation of our results. The *p*-values decrease as datasets and measurement precision increase (Wasserstein and Lazar, 2016), and the analysis of many coefficients increases the probability of type I errors (i.e. "false positives"; Oxman and Guyatt, 1992; Ioannidis, 2005). Further, the *p*-value is often misinterpreted (Goodman, 2008) and biased (Ioannidis, 2019), which may lead to overinterpretation of scientific results. We have accounted for these relations and possible overinterpretations by choosing a lower significance level than commonly used in ecology studies and complementing the *p*-value with the minimum Bayes factor of the null hypothesis to the alternative hypothesis. While ecological studies generally apply a significance level of $\alpha = 0.05$, we applied a smaller level of $\alpha = 0.01$ and thus a threshold of $p = 0.005$ for two-sided distributions. Even with this lower $\alpha$, most coefficients in our generalized additive model fits and ANOVAs were significant. Further, most of our statistically significant findings were accompanied by a minimum Bayes factor of 1/1000 or lower, indicating that our data suggest with decisive evidence that "no effect" is unlikely (i.e. the null hypothesis is rejected). As our study was exploratory, we believe that further studies to verify or falsify and quantify the presumed effects are worthwhile and will provide exciting new insights.

## 5 Conclusion

Based on our combined results, we recommend (1) species-specific models for the analysis of underlying phenology processes and for projections, (2) a combination of samples for cross-validation and independent test samples, and (3) to consider a possible tendency to the mean underlying the models. The choice of species- rather than site-specific models leads to generally larger sample sizes and larger ranges of considered drivers. In addition, species-specific models facilitate the assessment of the extent to which calibrated models tend towards the mean observation due to insufficient consideration of relevant processes. We advocate cross-validation of possibly regional, species-specific models, followed by independent tests. Specifically, we propose that (1) sites are selected in a stratified procedure based on annual mean temperature for (2) the cross-validation of a species-specific model with systematically balanced observations selected based on site and year, before (3) the calibrated

model is tested with new sites selected in a stratified procedure based on phenology. For both cross-validation and testing, the degree to which the model tends to the mean should be examined to assess how well the models perform at individual sites, within the entire region of interest, and, where possible, under different climate regimes.

We conclude that generally uncertain projections tend towards later autumn leaf phenology. Accurate projections of changes in the timing of autumn phenology under projected climate change are essential for our understanding of the future $CO_2$ mitigating capacity and of future species compositions and distributions of forests. Our results suggest that projected autumn phenology will be delayed due to projected climate warming, and thus the projected length of the growing season will increase. However, this result appears to be based on models that respond to quite similar underlying processes and may underestimate adverse effects of climate warming on autumn phenology, such as severe droughts. Therefore, further studies are needed to develop models that adequately account for processes relevant to autumn leaf phenology, and thus provide more valid projections.

**Code and data availability**

All manuscript related raw data are publicly available, properly referenced, and listed in Supplement S1: Table S7. The modelled and projected autumn phenology data is openly accessible under https://doi.org/10.5061/dryad.dv41ns22k and https://doi.org/10.5061/dryad.mw6m90613, respectively. In addition, the manuscript related R code for the phenology models is openly accessible under https://doi.org/10.5281/zenodo.7188160.

**Author contribution**

This study was conceptualized by MM and CB. MM adapted the code of the process-oriented models and performed the simulations. MM analyzed the simulations and visualized the results with contributions from CB. The manuscript was prepared by MM with contributions from CB.

**Competing interests**

The authors declare that they have no conflict of interest.

**Acknowledgements**

We thank Deborah Zani and Constantin Zohner for the informative discussions about their study (Zani et al., 2020) and the applied code for the phenology models (Zanid90, 2021), which we adapted for our study. We further thank Lidong Mo and Constantin Zohner from the Crowther Lab at ETH Zurich to share their code to download climate data from the Google Earth Engine (Gorelick et al., 2017), which we adapted for our study. We also thank Hussain Abbas from the Forest Ecology group

of ETH Zurich for his helpful advice and support with the batch system and the management of the computational jobs. Finally, we thank Nicolas Delpierre and the anonymous reviewer for their careful reviews and constructive suggestions, as well as the handling topic editor, Hans Verbeeck, for his invested time. Calibrations and projections of the phenology models, fits of the generalized additive models and subsequent analyses of variance were performed on the Euler cluster operated by the High Performance Computing group at ETH Zurich. This study is part of the project "Assessing climate influences on tree phenology and growth rates" (PheGro-Clim) at ETH Zurich, which was funded by the Swiss National Science Foundation SNSF (SNSF project number 179144).

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
