# Peer review of "Process-oriented models of autumn leaf phenology: ways to sound calibration and implications of uncertain projections"

_EGUsphere, 2022_

## Author Comment (AC2)

**Authors' answer (AC) to comments of second referee (RC2)**

RC2: Autumn leaf phenology impacts the biochemical and biophysical feedback of forests to climate. Modelling and projecting autumn leaf phenology of deciduous trees is therefore important and timely. Several studies have proposed and compared various modelling approaches. This study is different in the way that does not focus on a new modelling approach or only comparing existing approaches, but integrate model comparison with an analyze of the impact of different calibration procedures (e.g. site vs species), optimization, data sampling procedure etc considering their impact on model performance and model projections. For the latter aspects, analyses of the different scenarios is also considered. I find the study important and well done. The manuscript is also easy to read and very nicely synthesizes an huge amount of data. Practical useful recommendation are made in conclusions. I have however, some suggestions for improvement.

AC: Thank you for your nice summary of our study. We are happy to hear that you liked the manuscript. Your suggestions for improvement were duly considered and answered here below. Corresponding changes in the manuscript (following after your comments and our answers) are marked in green, whereas yellow and blue signify the changes in response to the comments of the 1st referee and other changes of the authors, respectively.

RC2: 1.while the text is clear, a scheme of the Methodology, thus a schematic synthesis of the different analyses performed, performance indicators used, etc, would be useful.

AC: Done.

*This Figure 2 is:*

[Figure]

**Figure 2: Concept and methods applied.** This study assumed that, in addition to phenology models and climate scenarios, the choice of optimization algorithm and calibration sample (i.e., sampling procedure and sample size) affect model performance and model projections (i.e., the root mean square error, RMSE, and the shift between autumn leaf phenology in 2080–2099 and 1980–1999, $\Delta_{100}$, respectively). To answer research questions I and II (RQ I & II), the effects of these factors on the RMSE and

Δ100 were quantified with linear mixed-effects models. Subsequently, the relative influence of the factors (e.g., all phenology models) on the explained variance ($\sigma^2$) of RMSE and $\Delta_{100}$ were quantified with type-III ANOVA. To answer RQ III, the effects on the RMSE were related to the effects on Δ100 and the influences on $\sigma^2_{\Delta_{100}}$ by calculating the Kendall rank correlations (e.g., between the effects of the phenology models on the RMSE and $\Delta_{100}$ or between the effect of the phenology models on the RMSE and the influence of each model on $\sigma^2_{\Delta_{100}}$. The phenology models were calibrated site- and species-specifically (i.e., one set of parameters per site and species vs. one set of parameters per species, respectively). Sample size was quantified by the number of observations relative to the number of free parameters in the phenology model (N:q), the average number of observations relative to the number of free parameters ($\overline{N}$:q), and the number of sites relative to the 500 sites of the entire population (s:S).

RC2: 2.I realize the analysis of the different formulation of the models considered is not the main focus of the study; yet, the different models are discussed and they will sure attract interest. So, I would add in Methods (not only in supplementary) a paragraph with a general description of the different type of model used (e.g. only driven by current temperature and photoperiod, or modulated by summer conditions, or by budburst timing), their key drivers etc. In practice, a description of Table 1.

AC: Done.

*Thus, the 2$^{nd}$ paragraph of Sect. 2.2 now reads:*

While all models differ in their functions and drivers considered, they can be grouped according to the formulation of the response curve of the senescence rate and of the threshold function (Table 1). Models within a particular group differ by the number of free parameters, by the determination of the initial day of the accumulation of the senescence rate, or by the seasonal drivers of the threshold. The difference in the number of free parameters is relevant for the groups Mon− (Co) and Mon+ (Co). These groups contain two models each, which differ by the two exponents for the effects of cooler and shorter days on the senescence rate. Each of these exponents can be calibrated to the values 0, 1, or 2 in the models with more parameters, whereas the exponents are set to 1 in the models with fewer parameters. The initial day of the accumulation of the senescence rate is either defined according to temperature or day length in the two models of the group Sig (Co). The one or two seasonal drivers considered by the models of the groups Mon− (Li), Mon+ (Li), and Sig (Li) are site-specific anomalies of the timing of spring phenology, the growing season index, and daytime net photosynthesis accumulated during the growing season ignoring or considering water limitation constraints, as well as the actual leafy season or growing season mean temperature, the low precipitation index averaged over the leafy season, or the adapted low precipitation index of the growing season. All models are explained in detail in Supplement S2).

RC2: 3.in Abstract and the entire text, I would not stress too much the modelled data of growing season length, rather focus on the date of autumn phenology. In fact, the data on growing season length are crucially affected by the spring phenology, which was only very coarsely estimated here.

AC: We deleted several references to the changes in the growing season.

RC2: 4.the authors does not consider in fully another source of uncertainty, which is the quality of the observational data, comprising past climate data. For example, is the biases associated with considering climate at 25 km resolution negligible? (L79) I'm worried particularly for larix sites, which are often found on mountain regions. Similarly: what about the spatial match between LAI and soil water characteristics used when compared to data on phenology from PEP? Could large biases (at site level) be introduced?

> AC: The resolution is coarse when it comes to simulate leaf phenology of a couple of trees at a particular site. We discuss this now in sect. 4.5.1 (L1048-1057). While there are finer gridded datasets available, the finer grid does not necessarily make the data more accurate. Alternatively, one may bias-correct and interpolate the data oneself. However, without meteorological measurements at the site of interest, one can only make sure, that the past and future data match, i.e., are equally inaccurate. Because this already increases the accuracy of projections, it is certainly a necessity when this accuracy is assessed. The main interest of our study, however, was to identify the relative importance of choices made during calibration for the resulting model performance and projections. This relative importance should remain largely unaffected by the degree of accuracy of the input data.

> *The 2nd paragraph of Sect. 4.5.1 reads:*
> Spatial and elevational differences between a particular site and the centre of the corresponding grid cell, from which the meteorological data were extracted, affect the input data. Gridded data may poorly represent the conditions at a particular site due to spatial and elevational differences. For example, precipitation and temperature can change in response to different terrain and the lapse rate, respectively, while the leaf area index and plant-available water capacity can change due to different vegetation and soil conditions. These effects of spatial and elevational differences were not considered in this study and may have led to inaccurate input data (e.g., average MAT for the site Grossarl, 47.2° N / 13.2° E at 900 m a.s.l., in the Austrian Alps was ~0.6° C, which makes beech growth unlikely; Holtmeier and Broll, 2020). The degree of inaccuracy probably differs between sites, which inflated the site effects on model performance and model projections. In contrast, the effects of models, sampling procedures, and optimization algorithms were probably unaffected by the inaccurate input data (cf. above), so these data most likely had a neglectable effect on our results.

RC2: 5.autumn leaf phenology is actually made up by several phenological events (e.g. onset of chlorophyll degradation, 50% leaf coloration, leaf fall), with timing varying of several weeks (e.g. Marien et al 2019 New Phytologist, doi: 10.1111/nph.15991); are the models simulating the same exact event? (which one?)

> AC: We applied the models to simulate BBCH94, defined as "40% of the leaves have colored or fallen" (Hack et al. 1992, Meier 2001) or "leaf colouration" (http://www.pep725.eu/; accessed on April 13, 2022). In their original publications, the models were used to simulate:
> - "leaf fall / yellowing" (Dufrêne et al. 2005)
> - "90% of the trees show yellow leaves over 20–50% of their crowns" (Delpierre et al. 2009)
> - "more than 50% of leaves have changed color" (Keenan and Richardson 2015)
> - "the day when almost all green leaves have colored" (Liu et al. 2019)
> - "the day when about 5% of canopy leaves turn from green to yellow or red on more than half of the observed trees" (Lang et al. 2019)

- "the date when 50% of leaves had lost their green color (BBCH94) or had fallen (BBCH95)" (Zani et al. 2020)

RC2: L164: to my knowledge, beech does not growth at site with MAT below 6-7 degree C. A beech site at 0.6 degree MAT (subarctic conditions) is quite unrealistic.

AC: We agree and interpret this outlier as a consequence of inaccurate weather data due to spatial and elevational differences between a particular site and the center of the corresponding grid cell. Thus, we now discuss this inaccuracy in sect. 4.5.1, where we also mentioned this example (see 2nd paragraph of Sect. 4.5.1 inserted further above).

RC2: L843-845: the explanation based on severity of extreme is questionable; see Marien et al 2021 Biogeosciences (doi.org/10.5194/bg-18-3309-2021), and for a more fundamental impact of drought on autumn phenology see Marchin et al 2010 Oecologie (DOI 10.1007/s00442-010-1614-4).

AC: As we understand Mariën et al. (2021), there is an important difference between an observation of autumn leaf phenology based on canopy greenness vs. chlorophyll content when it comes to discuss the effect of drought. In our study, we worked with observations of canopy greenness. Therefore, we believe that our explanation holds if we specify that we talk about canopy greenness.

*Thus, the 1st paragraph of Sect. 4.4 now reads:*
Overall, the climate projection scenarios were the primary drivers of the projected shifts in autumn phenology, with the warmer scenario causing later autumn phenology than the cooler scenario, which is consistent with the currently observed main effect of climate warming. Having the largest influence in two out of three projection modes, climate projection scenarios explained between 46% and 64% of the variance in the 100-year shifts of autumn phenology. On average, the projected autumn phenology occurred 8–9 days later when projected with the warmer RCP 8.5 than with the cooler RCP 4.5 scenarios, which corresponds to the observed main effect of warming. Past climate warming was found to mainly delay autumn phenology (Ibáñez et al., 2010; Meier et al., 2021), but slight forward shifts or a distribution around a temporal change rate of zero have also been observed (Menzel et al., 2020; Piao et al., 2019). Such inconsistent past trends may be explained by the fact that autumn phenology (i.e., observed with canopy greenness rather than chlorophyl content; cf. Sect. 2.1.1 and Mariën et al., 2021) depends more on the severity than the type of weather event, with, for example, moderate heat spells causing backward shifts but extreme heat spells and drought causing forward shifts (Xie et al., 2015). Since the number and severity of heat spells is related to sites (e.g. warmer lowland vs. cooler highland sites; Bigler and Vitasse, 2021), such opposing effects of weather events may explain the large influence of sites on projected shifts in autumn phenology, as discussed below. In addition, the length of the growing season is affected by shifts in spring and autumn phenology for deciduous trees. Our projections were based on spring phenology that advanced by 20 days within 100 years. Subsequently, the projected growing season lengthened by 7–32 days (RCP 4.5) or by 16–40 days (RCP 8.5), even when autumn phenology shifted forward, as projected with some models and discussed further below. Therefore, our study supports a general lengthening of the growing season due to projected climate warming, as also suggested by Delpierre et al. (2009), Keenan and Richardson (2015), and Meier et al. (2021), in contrast to Zani et al. (2020).

RC2: L906-907: "… all analyzed models are based on the same process …". I do not agree: models based on current autumn conditions (temperature and daylength) are different than models considering also the impact of, for example, summer (e.g. implying legacy of tree growth on senescence) or budburst (e.g. implying constraint on leaf longevity).

AC: We altered our conclusion slightly (L1016-L1017), while remaining convinced that effects other than temperature and day length remain under-considered by current models.

*Thus, the 4ᵗʰ paragraph of Sect. 4.4 now reads:*
The influence of phenology models on projected autumn phenology was relatively low and the range of projections relatively small. The largest influence of phenology models was 11% and occurred in projections based on site-specific models and hence was almost six times smaller than the influence of climate projection scenarios. While the underlying processes differ between each model (Delpierre et al., 2009; Keenan and Richardson, 2015; Lang et al., 2019; Liu et al., 2019; Zani et al., 2020), the influence of these differences on the projected autumn phenology did not affect the projected lengthening of the growing season: Different models altered the reference shifts of +8.2 to +11.6 days by -12 to +2 days, which resulted in some forward shifts in autumn phenology with the cooler RCP 4.5 scenarios, but never in a shortening of the growing season because the latter is calculated in combination with the -20 days shift in spring phenology. Moreover, the difference between the models lay within 14 days (i.e., −12 to +2 days), which is less than the uncertainty attached to recordings of autumn phenology based on human observations (i.e. due to small sample sizes and observer bias; Liu et al., 2021). In other words, the different process-oriented models led to differences in the length of the growing season that were smaller than the uncertainty in the data upon which we based our projections. Therefore, our results justify the assumption, that the examined phenology models do not differ fundamentally in their underlying processes, even if we acknowledge that the TDM, PDM, and TPDM models (Liu et al., 2019) behaved differently than the other models (i.e. they resulted in the largest forward or smallest backward shifts of autumn phenology). Rather, we suggest that the effects of temperature and day length, which all analyzed models simplify in different ways, mostly suppress the effects of other concerned drivers.

**References of authors' answer:**

Delpierre, N., E. Dufrene, K. Soudani, E. Ulrich, S. Cecchini, J. Boe, and C. Francois. 2009. Modelling interannual and spatial variability of leaf senescence for three deciduous tree species in France. Agricultural and Forest Meteorology **149**:938-948.

Dufrêne, E., H. Davi, C. Francois, G. le Maire, V. Le Dantec, and A. Granier. 2005. Modelling carbon and water cycles in a beech forest Part I: Model description and uncertainty analysis on modelled NEE. Ecological Modelling **185**:407-436.

Hack, H., H. Bleiholder, L. Buhr, U. Meier, U. Schnock-Fricke, E. Weber, and A. Witzenberger. 1992. Einheitliche Codierung der phänologischen Entwicklungsstadien mono-und dikotyler Pflanzen – Erweiterte BBCH-Skala, Allgemein. Nachrichtenbl. Deut. Pflanzenschutzd **44**:265-270.

Keenan, T. F., and A. D. Richardson. 2015. The timing of autumn senescence is affected by the timing of spring phenology: implications for predictive models. Glob Chang Biol **21**:2634-2641.

Lang, W., X. Chen, S. Qian, G. Liu, and S. Piao. 2019. A new process-based model for predicting autumn phenology: How is leaf senescence controlled by photoperiod and temperature coupling? Agricultural and Forest Meteorology **268**:124-135.

Liu, G., X. Q. Chen, Y. S. Fu, and N. Delpierre. 2019. Modelling leaf coloration dates over temperate China by considering effects of leafy season climate. Ecological Modelling **394**:34-43.

Meier, U. 2001. Growth stages of mono-and dicotyledonous plants. 2. Edition edition. Blackwell Wissenschafts-Verlag.

Zani, D., T. W. Crowther, L. Mo, S. S. Renner, and C. M. Zohner. 2020. Increased growing-season productivity drives earlier autumn leaf senescence in temperate trees. Science **370**:1066-1071.

---

## Author Response (AR1)

**Author's answer (AC) to comments of first referee (RC1)**

*General comments:*

RC1: This manuscript reports a very comprehensive assesment of current process-oriented models of leaf senescence in temperate deciduous trees. It considers aspects that are rarely addressed in phenological modelling, and often overlooked or reported quite superficially in other manuscripts, despite their potential strong influence on the interpretation of the research. The considered aspects are related to the model calibration and evaluation (namely the scale of calibration site- vs. species-scale, the choice and parameterization of optimization algorithms, the cal/val sampling strategy), and their effect on model projections.

In essence, this manuscript has the potential to become a vademecum for phenological modellers, and possibly beyond this community (i'm thinking here of modellers working with models simple/fast enough to allow large numbers of computations), providing an example on how to rigorously design cal/val and projection studies. The downside is that the manuscript is very long, and sometimes difficult to follow due to the comprehensiveness of the tests performed and the results reported. This is not prohibitive to me, and I would like to read more often phenological modelling studies conducted in such a rigorous way. Hence I do not ask for a general reduction of the manuscript length. However, I strongly recommend the authors to provide a section (such sections are called "boxes" in some journals) highlighting what they identified from their work as good practices for phenological modelling.

This section would ideally list items regarding to the aspects they deal with (e.g. "how to rigorously sample a phenological database for cal/val of a phenological model", "which optimization algorithm to choose" etc.) and giving practical numbers / orders of magnitude / rules of thumbs useful to other modellers. This may require including in this "box" some definitions located here and elsewhere in the manuscript, possibly with examples (e.g. what is a "stratified" sampling etc.). I think most of this is already present in the manuscript, but it is dispersed and quite difficult to find. To put it more bluntly, currently the manuscript has a strong potential but no strong take-home message, and leaves the reader wrung out after an avalanche of valuable informations.

In other words, I would like this manuscript to offer two levels of reading: the very detailed one that is currenctly presented. And another, more synthetic one, which would help spread good practices in phenological modelling.

> AC: Thank you for your nice summary and general comments. We are happy to hear that you see the potential of our manuscript becoming a vademecum. We considered your suggestion of adding a section/box that synthesizes our methods and results as well as provides practical guidelines to modelers. Further, your specific comments were duly considered and answered here below. Corresponding changes in the marked-up version of our revised manuscript are highlighted in yellow, whereas green and blue signify the changes in response to the comments of the 2$^{nd}$ referee and other changes of the authors, respectively. Line numbers refer to the marked-up version of our revised manuscript.

*Specific comments:*

RC1: L25-26: should this sentence be understood "in general" for all leaf senescence models?

> AC: Yes. We made this clearer with the expression "current models". (L28)

RC1: L50: In French, it is customary to call this scientist "Réaumur" (not "De Réaumur")

  AC: We changed this accordingly. (L116)

RC1: L63: meaning of "was not given" ? Clarify

  AC: We clarified the sentence. (L127-128)

RC1: L88-89: about space-for-time approach, the paper by Jochner et al. is interesting, and not appearing in the reference list: Susanne Jochner, Amelia Caffarra, Annette Menzel, Can spatial data substitute temporal data in phenological modelling? A survey using birch flowering, Tree Physiology, Volume 33, Issue 12, December 2013, Pages 1256–1268, https://doi.org/10.1093/treephys/tpt079

  AC: We were not aware of this publication and cited it now. Of course, the impossibility of site-specific models to project to new sites remains, why we did not alter our statement but referenced the study you mentioned above. (L155)

RC1: L113 "in contrast": I do not understand the logical link with preceding sentence here

  AC: We clarified the sentence. (L179-180)

RC1: L120-121: site-specific vs. species-specific calibration: behind this is the question of tree populations local adaptations, that is not considered here. It was in Delpierre et al. 2009 (and Chuine et al. 2000). Mention it somewhere.

  AC: We briefly mentioned local adaption and the indicated references in L150-151 and L153-154.

RC1: L153: the assertion "the proper order (e.g. the date for leaf coloration was before the date for leaf fall)" is wrong: leaf fall can occur before leaf coloration. Or at least, part of the leaf fall can occur before reaching BBCH94 (= 40% of leaves colored or fallen) considered in this paper (L159). Which BBCH code did you consider for leaf fall?

  AC: We checked the order according to the BBCH codes, i.e., BBCH10 had to occur before BBCH11 (i.e., leaf separation before leaf unfolding) and BBCH94 had to occur before BBCH95 (i.e., 40% of leaves colored or fallen before 50% of leaves colored or fallen). To our knowledge, these orders cannot be reversed. However, the description of the codes was certainly misleading. We changed this accordingly in L219-220 and L225-226.

RC1: L154: "After corresponding correction..." is unclear. Rephrase.

  AC: We clarified the sentence. (L221)

RC1: L174: why using Tmin as a "general" driver? Models often use the daily average temperature.

  AC: Indeed, except for the models TPMt, TPMp, PIAGSI, PIA+ and PIA- (Lang et al. 2019, Zani et al. 2020), all compared models use daily average temperature, whereas Jibran (2013) and Lim et al. (2007) describe cold stress to promote leaf senescence.

Since cold stress likely relates more to daily minimum temperature, we used the latter in all models. This is also what was done in the currently most recent model comparison (Zani et al. 2020), which can now directly be compared with our comparison. However, we agree that this adaptation of the models CDD, DM1, DM2, SIAM, TDM1, TDM2, PDM1, PDM2, and TPDM1, and TPDM2 must be pointed out and discussed, both of which we did in sections 2.2 and 4.5.2 (L298-299 and Sect. 4.5.2, L1058-1068).

RC1: L178-179: 0.25° is a quite coarse spatial resolution, notably when it comes to mountainous areas. Any correction of temperature with altitude (through lapse rate)?

> AC: To not further complicate and lengthen the study, we did not correct temperature with elevation. However, we discussed this in the newly added section 4.5.1 (Sect. 4.5.1, L1034-1057).

RC1: L183: what is a "climate model chain"? Is one CMC corresponding to one particular climate model run under a particular RCP?

> AC: A CMC is a particular combination of a global and regional climate model. (Clarified in L249-250)

RC1: Table 1: From Suppl Mat 2, it is unclear how you implemented the relatively complex responses to GSI and Anet described in Ziani et al. 2020. Did you actually code those responses? What cast doubts to me is your use of the term "apparent photosynthesis" in Suppl Mat 2, though it seems from Suppl Mat 1 that you indeed computed photosynthesis. This should be clarified in Suppl Mat 2, e.g. with a mention to Ziani et al. 2020 (their suppl. mat.).

> AC: Yes, we coded these responses according to the Supplement S3 (Sect. S2.2 and S2.3). Following your comment, we clarified that GSI, Anet and Anet–w were calculated by ourselves. (Supplement S2: p. 2 and Supplement S3: pp. 1 and 4) Unfortunately, we are unable to fully understand your doubts as "Suppl Mat" 1 and 2 probably point to Supplements S1 and S2 but these supplements explain the used data (S1) and phenology models (S2). However, we believe you referred to the Supplements S2 and S3 instead, which explain the used models and model drivers, respectively, and answer your comment accordingly. To our understanding of Egle (1960) and Wohlfahrt and Gu (2015), for example, the apparent photosynthesis ($A_{net}$) equals the difference between the real (or gross) photosynthesis ($A_{gd}$) and light (or daytime) respiration. And this is what we calculated in Eq. S32 of Supplement S3, defining light respirations as the daytime fraction of apparent respiration ($R_d$): $A_{net} = A_{gd} - R_d \times L / 24$.
> Our understanding from Zani et al. (2020) is that they also did the following. While they described their models in the main part as dependent on photosynthesis, Eqs. 20 and 22 in their supplement clearly point to the apparent photosynthesis.
> Therefore, we strongly agree with you that it must be clarified if the apparent or real photosynthesis was used. We now have done so in the changes in section S2.3 of Supplement S3. In addition, we calculated these drivers ourselves, which we also specified in section S2 of Supplement S3.

RC1: L248: were the models tested in their ability to simulate trends in observed data (if any over 20-65 years?)

AC: Model accuracy was assessed solely based on the RMSE and we did not perform an additional assessment of the model's ability to simulate past trends. However, this would certainly be an interesting study, which we may conduct in the future.

RC1: L259-260: Does this mean that the parameters used for model evaluation and model projections were possibly different? Why so?

AC: Yes, in site-specific calibration, the parameters were possibly different. The site-specific models were evaluated with a 5-fold cross-validation, which leads to five sets of parameters. Which of these sets should be used for projections? Maybe the set that led to the smallest RMSE or the average parameters calculated from all five sets? On the one hand, different RMSE are likely related to differences in the observations used for validation and thus do not justify the use of one parameter set over another. On the other hand, average parameters are unlikely to lie in any local optimum or even the global optimum. Compared to these alternatives, the selection of parameters derived from the calibration with the whole data appeared most reliable to us.

RC1: L440-441: I'm unsure we are here talking about model external validation. If this is the case, it is not particularly surprising that site-specific calibration can sometimes yield, at the very same site, unrealistic results when the model is used to predict unknown data. Indeed, site-specific parametrisation are more prone to over-fitting (few data points over which to fit the model), as compared to species-specific parametrisation (which includes many sites).

AC: We are not sure, if we understand you correctly. With "external validation" we refer to the fact, that the models were validated with observations that have not yet been used for their calibration. These "external" observations, however, were recorded at the same site as the observations used for calibration. This may lead to unrealistic results due to overfitting, if the number of observations left for calibration is low, which is exactly what we wanted to study (i.e., effects of the sample size). We agree that such unrealistic results should be expected, why we deleted the word "surprisingly" in L440 (now L535).

RC1: Fig. 3b: inversion of the x-axis is confusing. It took me several minutes to understand this subplot (as compared to the text description), because i intuitively tried to interpret the x-axis with negative values on the left of the vertical dashed line.

AC: We felt that the plot is easier to understand when more accurate models are visualized with a dot further to the right, but maybe we were wrong. Since we cannot be sure, what our readers will prefer, we changed the plots such that the x-axes are not inversed anymore (L581).

RC1: L531: how possible is this? Considering that NA-producing and non-converging runs are assigned high RMSE values.

AC: We agree that this finding for species-specific models is counterintuitive. However, on the one hand, the LMMs are not perfect (adjusted $R^2$ range from 0.41 to 0.54) and the ranks of the reference model CDD, whose RMSE is specified on L618 and which has confused you, changed from 14 to 18 and 14 for species-specific models validated within sample and population, respectively. On the other hand, large samples led to most NA-containing calibration runs (Supplement S6: Fig. S1). At the same time, GenSA large samples also led to the lowest RMSE when NA-values were

replaced with 170 d. It may well be that many of these NA-producing runs based on large samples led to lower-than-average external RMSEs, especially since the punishing effect of a particular replacement is weaker in large samples. We added a paragraph in section 4.3 (L930-942), in which we discuss this issue.

RC1: L534-548: a question relative to optimization algorithm is: is the identity of the algorithm involved, or is it more the design of the simulation: in other words we see in SM4, Table S1 that the "normal" vs. "extended" runs of the calibration procedures include less vs. more iterations. I do not see an analysis considering the influence of this number of iterations per algorithm. Apparently, the influence of iteration number is small (Fig 3b: points are close whether considering "norm" or "extd" for one algorithm). If the "extd" simulation number was extended further, would that modify the results?

> AC: The influence of the number of iterations per algorithm can be seen in Fig. 3a, for example. More iterations led to better results in most cases except for CMA-ES in site-specific models and for PSO in species-specific models. Since the number of iterations affects the step size with which the global optimum is searched (exploration-exploitation trade-off; L875), we assume that the step size became too small in the afore mentioned exceptions. This led us to the conclusion that careful tuning of the algorithms is very important (L875-878).
> We agree that there was no specific analysis of the influence of this number. Analyzing the effect of only two states, i.e., "few" vs. "many" iterations is certainly not enough. A profound study of the effect of the number of iterations together with a suggestion of how to set the corresponding parameters of the algorithm would be helpful for the modelers.

RC1: Fig 4b: if CMC numbers point to particular climate models, one sees the effect of the climate model can be huge ! Were these climate models unbiased (against local observed climate data)? Climate model bias can influence process-based model simulations strongly, see e.g. Jourdan et al. 2021

> AC: Yes, CMC numbers point to particular models. For time reasons, we refrained from bias correcting these data, but we now discuss this in section 4.5.1 (L1034-L1058).

RC1: L621-623: I do not see this on Fig. 4b (i.e. dot coefficients of DM2 and DM2Za20 are not remarkable relative to other models)

> AC: Yes. Here we refer to Supplement S6: Fig. S6; Supplement S6: Table S27 (L718).

RC1: L698: rewrite to "phenology"

> AC: Done (now L793)

RC1: L711-712: recall which models lead to the best results in Zani et al. 2020

> AC: We already discuss the best models in sect. 4.1 and mention the corresponding original studies. We now referred to Lang et al. (2019) and Zani et al. (2020) as studies with which our findings do not agree (L834-836).

RC1: L741 "other models": recall briefly their characteristics here

AC: Done. See L834-836.

RC1: L750: rephrase to "local adaptation (Peaucelle et al. 2019)"

AC: Done. See L844.

RC1: L752: "the less such consideration is possible" : unclear, rephrase.

AC: Done. See L848-850.

RC1: L758-759: and/or that the observed data are more prone to observation bias, that can magnify if same observer is operating across years at a given site (in practice, observers inter-calib are rare). See Liu et al. 2021

AC: We added corresponding thoughts to Sect. 4.1 (L845-846, L855-856).

RC1: L788-789: interesting result. Where does this "17 sites" come from? I do not remember seeing that earlier in the manuscript.

AC: 17 is the number of sites in stratified samples based on the average timing of autumn phenology (L348). The result is mentioned in L549–L560, visualized in Figure 3d, and quantified in Supplement 6: Table S2.

RC1: L819: Cochran (1946) is missing from the reference list

AC: Done. See L1230-1231.

RC1: L852: remove "but see"

AC: Done. See L961-962.

RC1: L889-891: we touch here the question of local adaptation again

AC: Yes. We now mentioned this explicitly in L991.

RC1: L897: "Different models altered the reference shifts by -12 to +2 days", recall the average.

AC: Done. See L1007.

RC1: L970: "... and found our data to strongly encourage further research" is unclear.

AC: We tried to refer to the exploratory nature of our study. Since we mentioned this in L1125, we simply deleted it here (L1115).

**Author's answer (AC) to comments of second referee (RC2)**

RC2: Autumn leaf phenology impacts the biochemical and biophysical feedback of forests to climate. Modelling and projecting autumn leaf phenology of deciduous trees is therefore important and timely. Several studies have proposed and compared various modelling approaches. This study is different in the way that does not focus on a new modelling approach or only comparing existing approaches, but integrate model comparison with an analyze of the impact of different calibration procedures (e.g. site vs species), optimization, data sampling procedure etc considering their impact on model performance and model projections. For the latter aspects, analyses of the different scenarios is also considered. I find the study important and well done. The manuscript is also easy to read and very nicely synthesizes an huge amount of data. Practical useful recommendation are made in conclusions. I have however, some suggestions for improvement.

> AC: Thank you for your nice summary of our study. We are happy to hear that you liked the manuscript. Your suggestions for improvement were duly considered and answered here below. Corresponding changes in the marked-up version of our revised manuscript are highlighted in green, whereas yellow and blue signify the changes in response to the comments of the 1ˢᵗ referee and other changes of the authors, respectively. Line numbers refer to the marked-up version of our revised manuscript.

RC2: 1.while the text is clear, a scheme of the Methodology, thus a schematic synthesis of the different analyses performed, performance indicators used, etc, would be useful.

> AC: Done (Fig. 2, L484-495).

RC2: 2.I realize the analysis of the different formulation of the models considered is not the main focus of the study; yet, the different models are discussed and they will sure attract interest. So, I would add in Methods (not only in supplementary) a paragraph with a general description of the different type of model used (e.g. only driven by current temperature and photoperiod, or modulated by summer conditions, or by budburst timing), their key drivers etc. In practice, a description of Table 1.

> AC: Done (L273-L285).

RC2: 3.in Abstract and the entire text, I would not stress too much the modelled data of growing season length, rather focus on the date of autumn phenology. In fact, the data on growing season length are crucially affected by the spring phenology, which was only very coarsely estimated here.

> AC: We deleted several references to the changes in the growing season (L19, L20, L22, L1009, L1018, L1020, L1026, 1139 and L1140)

RC2: 4.the authors does not consider in fully another source of uncertainty, which is the quality of the observational data, comprising past climate data. For example, is the biases associated with considering climate at 25 km resolution negligible? (L79) I'm worried particularly for larix sites, which are often found on mountain regions. Similarly: what about the spatial match between LAI and soil water characteristics used when compared to data on phenology from PEP? Could large biases (at site level) be introduced?
* * *
AC: The resolution is coarse when it comes to simulate leaf phenology of a couple of trees at a particular site. We discuss this now in sect. 4.5.1 (L1048-1057). While there are finer gridded datasets available, the finer grid does not necessarily make the data more accurate. Alternatively, one may bias-correct and interpolate the data oneself. However, without meteorological measurements at the site of interest, one can only make sure, that the past and future data match, i.e., are equally inaccurate. Because this already increases the accuracy of projections, it is certainly a necessity when this accuracy is assessed. The main interest of our study, however, was to identify the relative importance of choices made during calibration for the resulting model performance and projections. This relative importance should remain largely unaffected by the degree of accuracy of the input data.

RC2: 5.autumn leaf phenology is actually made up by several phenological events (e.g. onset of chlorophyll degradation, 50% leaf coloration, leaf fall), with timing varying of several weeks (e.g. Marien et al 2019 New Phytologist, doi: 10.1111/nph.15991); are the models simulating the same exact event? (which one?)

AC: We applied the models to simulate BBCH94 (L219-226), defined as "40% of the leaves have colored or fallen" (Hack et al. 1992, Meier 2001) or "leaf colouration" (http://www.pep725.eu/; accessed on April 13, 2022). In their original publications, the models were used to simulate:
- "leaf fall / yellowing" (Dufrêne et al. 2005)
- "90% of the trees show yellow leaves over 20–50% of their crowns" (Delpierre et al. 2009)
- "more than 50% of leaves have changed color" (Keenan and Richardson 2015)
- "the day when almost all green leaves have colored" (Liu et al. 2019)
- "the day when about 5% of canopy leaves turn from green to yellow or red on more than half of the observed trees" (Lang et al. 2019)
- "the date when 50% of leaves had lost their green color (BBCH94) or had fallen (BBCH95)" (Zani et al. 2020)

RC2: L164: to my knowledge, beech does not growth at site with MAT below 6-7 degree C. A beech site at 0.6 degree MAT (subarctic conditions) is quite unrealistic.

AC: We agree and interpret this outlier as a consequence of inaccurate weather data due to spatial and elevational differences between a particular site and the center of the corresponding grid cell. Thus, we now discuss this inaccuracy in sect. 4.5.1, where we also mentioned this example (L1053-1054).

RC2: L843-845: the explanation based on severity of extreme is questionable; see Marien et al 2021 Biogeosciences (doi.org/10.5194/bg-18-3309-2021), and for a more fundamental impact of drought on autumn phenology see Marchin et al 2010 Oecologie (DOI 10.1007/s00442-010-1614-4).

AC: As we understand Mariën et al. (2021), there is an important difference between an observation of autumn leaf phenology based on canopy greenness vs. chlorophyll content when it comes to discuss the effect of drought. In our study, we worked with observations of canopy greenness. Therefore, we believe that our explanation holds if we specify that we talk about canopy greenness (L952).

RC2: L906-907: "… all analyzed models are based on the same process …". I do not agree: models based on current autumn conditions (temperature and daylength) are different than models considering also the impact of, for example, summer (e.g. implying legacy of tree growth on senescence) or budburst (e.g. implying constraint on leaf longevity).

> AC: We altered our conclusion slightly (L1016-L1017), while remaining convinced that effects other than temperature and day length remain under-considered by current models.

**List of relevant changes**

| Change | As reaction to | Line number* |
|---|---|---|
| Insertion of summary (i.e., box) of our study | RC1 | 35 – 98 |
| New paragraph in Sect. 2.2 that briefly describes the used models | RC2 | 273 – 285 |
| New Fig. 2, which gives an overview of the applied methods | RC2 | 484 – 495 |
| Change of x-axis in Fig. 3b such that is not inverted anymore | RC1 | 581 |
| New paragraph in Sect. 4.3 to discuss the effect of NA substitution vs. the exclusion of NA-yielding runs on the RMSE | RC1 | 930 – 942 |
| New Sect. 4.5.1 to discuss uncertainty in driver data | RC1 and RC2 | 1034 – 1057 |

*) Line numbers refer to the marked-up version of our revised manuscript.

**References:**

Delpierre, N., E. Dufrene, K. Soudani, E. Ulrich, S. Cecchini, J. Boe, and C. Francois. 2009. Modelling interannual and spatial variability of leaf senescence for three deciduous tree species in France. Agricultural and Forest Meteorology **149**:938-948.

Dufrêne, E., H. Davi, C. Francois, G. le Maire, V. Le Dantec, and A. Granier. 2005. Modelling carbon and water cycles in a beech forest Part I: Model description and uncertainty analysis on modelled NEE. Ecological Modelling **185**:407-436.

Egle, K. 1960. Apparente und reelle Photosynthese, Gaswechselgleichgewicht, Lichtatmung. Pages 182-210 *in* A. Pirson, editor. Die CO2-Assimilation / The Assimilation of Carbon Dioxide: In 2 Teilen / 2 Parts. Springer Berlin Heidelberg, Berlin, Heidelberg.

Hack, H., H. Bleiholder, L. Buhr, U. Meier, U. Schnock-Fricke, E. Weber, and A. Witzenberger. 1992. Einheitliche Codierung der phänologischen Entwicklungsstadien mono-und dikotyler Pflanzen – Erweiterte BBCH-Skala, Allgemein. Nachrichtenbl. Deut. Pflanzenschutzd **44**:265-270.

Jibran, R., D. A. Hunter, and P. P. Dijkwel. 2013. Hormonal regulation of leaf senescence through integration of developmental and stress signals. Plant Molecular Biology **82**:547-561.

Keenan, T. F., and A. D. Richardson. 2015. The timing of autumn senescence is affected by the timing of spring phenology: implications for predictive models. Glob Chang Biol **21**:2634-2641.

Lang, W., X. Chen, S. Qian, G. Liu, and S. Piao. 2019. A new process-based model for predicting autumn phenology: How is leaf senescence controlled by photoperiod and temperature coupling? Agricultural and Forest Meteorology **268**:124-135.

Lim, P. O., H. J. Kim, and H. Gil Nam. 2007. Leaf senescence. Annual Review of Plant Biology **58**:115-136.

Liu, G., X. Q. Chen, Y. S. Fu, and N. Delpierre. 2019. Modelling leaf coloration dates over temperate China by considering effects of leafy season climate. Ecological Modelling **394**:34-43.

Meier, U. 2001. Growth stages of mono-and dicotyledonous plants. 2. Edition edition. Blackwell Wissenschafts-Verlag.

Wohlfahrt, G., and L. Gu. 2015. The many meanings of gross photosynthesis and their implication for photosynthesis research from leaf to globe. Plant, cell & environment **38**:2500-2507.

Zani, D., T. W. Crowther, L. Mo, S. S. Renner, and C. M. Zohner. 2020. Increased growing-season productivity drives earlier autumn leaf senescence in temperate trees. Science **370**:1066-1071.

---

## Referee Report (RR1)

**Authors' answer (AC) to comments of first referee (RC1)**

*General comments:*

RC1: This manuscript reports a very comprehensive assessment of current process-oriented models of leaf senescence in temperate deciduous trees. It considers aspects that are rarely addressed in phenological modelling, and often overlooked or reported quite superficially in other manuscripts, despite their potential strong influence on the interpretation of the research. The considered aspects are related to the model calibration and evaluation (namely the scale of calibration site- vs. species-scale, the choice and parameterization of optimization algorithms, the cal/val sampling strategy), and their effect on model projections.

In essence, this manuscript has the potential to become a vademecum for phenological modelers, and possibly beyond this community (I'm thinking here of modelers working with models simple/fast enough to allow large numbers of computations), providing an example on how to rigorously design cal/val and projection studies. The downside is that the manuscript is very long, and sometimes difficult to follow due to the comprehensiveness of the tests performed and the results reported. This is not prohibitive to me, and I would like to read more often phenological modelling studies conducted in such a rigorous way. Hence I do not ask for a general reduction of the manuscript length. However, I strongly recommend the authors to provide a section (such sections are called "boxes" in some journals) highlighting what they identified from their work as good practices for phenological modelling.

This section would ideally list items regarding to the aspects they deal with (e.g. "how to rigorously sample a phenological database for cal/val of a phenological model", "which optimization algorithm to choose" etc.) and giving practical numbers / orders of magnitude / rules of thumbs useful to other modelers. This may require including in this "box" some definitions located here and elsewhere in the manuscript, possibly with examples (e.g. what is a "stratified" sampling etc.). I think most of this is already present in the manuscript, but it is dispersed and quite difficult to find. To put it more bluntly, currently the manuscript has a strong potential but no strong take-home message, and leaves the reader wrung out after an avalanche of valuable information.

In other words, I would like this manuscript to offer two levels of reading: the very detailed one that is currently presented. And another, more synthetic one, which would help spread good practices in phenological modelling.

> AC: Thank you for your nice summary and general comments. We are happy to hear that you see the potential of our manuscript becoming a vademecum. We considered your suggestion of adding a section/box that synthesizes our methods and results as well as provides practical guidelines to modelers. Further, your specific comments were duly considered and answered here below. Corresponding changes in the manuscript (following after your comments and our answers) are marked in ==yellow==, whereas ==green== and ==blue== signify the changes in response to the comments of the 2$^{nd}$ referee and other changes of the authors, respectively.
>
> *We now inserted a summary (i.e., a box), which reads:*

[revised manuscript text omitted]

*Specific comments:*

RC1: L25-26: should this sentence be understood "in general" for all leaf senescence models?

    AC: Yes. We made this clearer with the expression "current models".

    *Thus, the 3$^{rd}$ paragraph of the abstract now reads:*
    Our results justify inferences from comparisons of process-oriented phenology models to phenology-driving processes and we advocate species-specific models for such analyses and subsequent projections. For sound calibration, we recommend a combination of cross-validations and independent tests, using randomly selected sites from stratified bins based on mean annual temperature and average autumn phenology, respectively. Poor performance and little influence of phenology models on autumn phenology projections suggest that current models are overlooking relevant drivers. While the uncertain projections indicate an extension of the growing season, further

studies are needed to develop models that adequately consider the relevant processes for autumn phenology.

RC1: L50: In French, it is customary to call this scientist "Réaumur" (not "De Réaumur")

AC: We changed this accordingly.

RC1: L63: meaning of "was not given" ? Clarify

AC: We clarified the sentence. (L127-128)

*Thus, the 4th paragraph of Sect. 1 now reads:*
Models of spring phenology regularly outcompete models of autumn phenology by several days when assessed by the root mean square error between observed and modelled dates (4–9 vs. 6–13 days, respectively; Basler, 2016; Liu et al., 2020). These errors have been interpreted in different ways and have multiple sources. Basler (2016) compared over 20 different models and model combinations for spring leaf phenology of trees. He concluded that the models underestimated the inter-annual variability of observed dates of spring leaf phenology and were not transferable between sites. Liu et al. (2020) compared six models of autumn leaf phenology of trees and concluded that the inter-annual variability was well represented by the models, while their representation of the inter-site variability was relatively poor.

RC1: L88-89: about space-for-time approach, the paper by Jochner et al. is interesting, and not appearing in the reference list: Susanne Jochner, Amelia Caffarra, Annette Menzel, Can spatial data substitute temporal data in phenological modelling? A survey using birch flowering, Tree Physiology, Volume 33, Issue 12, December 2013, Pages 1256–1268, https://doi.org/10.1093/treephys/tpt079

AC: We were not aware of this publication and cited it now. Of course, the impossibility of site-specific models to project to new sites remains, why we did not alter our statement but referenced the study you mentioned above.

RC1: L113 "in contrast": I do not understand the logical link with preceding sentence here

AC: We clarified the sentence.

*Thus, the 9th paragraph of Sect. 1 now reads:*
Sample size in terms of the number of observations per site and the number of sites may influence the quality of phenology models as well. Studies on phenology models have usually selected sites with at least 10 or 20 observations per site, independent of the calibration mode (e.g. Delpierre et al., 2009; Keenan and Richardson, 2015; Lang et al., 2019). In studies with species-specific models, a wide range in the number of sites have been considered, namely 8 to >800 sites (e.g. Liu et al., 2019; Liu et al., 2020). In site-specific calibration, the number of sites may be neglected as the site-specific models cannot be applied to other sites. However, the number of observations is crucial, as small samples may lead to overfitted models due to the bias-variance trade-off (James et al., 2017, Ch. 2.2.2), i.e. the trade-off between minimizing the prediction error in the validation sample versus the variance of the estimated parameters in the calibrated models. To our knowledge, no study to date has examined possible overfitting in phenology models. In addition, in species-specific calibration,

the number of sites could influence the degree to which the population is represented by the species-specific models. While such reasoning appears intuitively right, we are unaware of any study that has systematically researched the correlation between the number of sites and the degree of representativeness.

RC1: L120-121: site-specific vs. species-specific calibration: behind this is the question of tree populations local adaptations, that is not considered here. It was in Delpierre et al. 2009 (and Chuine et al. 2000). Mention it somewhere.

AC: We briefly mentioned local adaption and the indicated references.

*Thus, the 6th paragraph of Sect. 1 now reads:*
When considering phenology data from different sites, one must, in principle, decide between two calibration modes, namely a calibration per site and species or a calibration over various sites with pooled data per species. While the former calibration leads to a set of parameters per species and site, the latter leads to one set of parameters per species. On the one hand, site-specific models may respond to local adaptation (Chuine et al., 2000) without explicitly considering the underlying processes as well as to relevant but unconsidered drivers. For example, a model based solely on temperature may provide accurately modelled data due to site-specific thresholds, even if the phenological observations at some sites are driven by additional variables such as soil water balance. On the other hand, species-specific models may consider local adaptation via parameters such as day length (Delpierre et al., 2009) and may be better suited for projections to  other sites and changed climatic conditions, as they apply to the whole species and follow a space-for-time approach (but see Jochner et al., 2013).

RC1: L153: the assertion "the proper order (e.g. the date for leaf coloration was before the date for leaf fall)" is wrong: leaf fall can occur before leaf coloration. Or at least, part of the leaf fall can occur before reaching BBCH94 (= 40% of leaves colored or fallen) considered in this paper (L159). Which BBCH code did you consider for leaf fall?

AC: We checked the order according to the BBCH codes, i.e., BBCH10 had to occur before BBCH11 (i.e., leaf separation before leaf unfolding) and BBCH94 had to occur before BBCH95 (i.e., 40% of leaves colored or fallen before 50% of leaves colored or fallen). To our knowledge, these orders cannot be reversed. However, the description of the codes was certainly misleading. We changed this accordingly.

*Thus, the 1st paragraph of Sect. 2.1.1 now reads:*
We ran our computer experiment with leaf phenology observations from Central Europe for common beech (Fagus sylvatica L.), pedunculate oak (Quercus robur L.), and European larch (Larix decidua MILL.). All phenological data were derived from the PEP725 project database (http://www.pep725.eu/; accessed on April 13, 2022). The PEP725 dataset mainly comprises data from 1948–2015 that were predominantly collected in Austria, Belgium, Czech Republic, Germany, the Netherlands, Switzerland, and the United Kingdom (Templ et al., 2018). We only considered site-years for which the phenological data were in the proper order (i.e., the first leaves have separated before they unfolded, BBCH10 before BBCH11, and 40% of the leaves have colored or fallen before 50% of the leaves, BBCH94 before BBCH95; Hack et al., 1992; Meier, 2001) and the period between spring and autumn phenology was at least 30 days. Subsequently, we only considered sites with at least 20 years for which

both spring and autumn phenology data were available. We randomly selected 500 of these sites per species. Each of these sites comprised 20–65 (beech), 20–64 (oak), or 20–30 (larch) site-years, all of which included a datum for spring and autumn phenology. This added up to 17 211 site-years for beech, 16 954 site-years for oak, and 11 602 site-years for larch. Spring phenology corresponded to BBCH11 for beech and oak and BBCH10 for larch, while autumn phenology for all three species was represented by ==BBCH94, hence forward referred to as leaf coloration== (Hack et al., 1992; Meier, 2001).

RC1: L154: "After corresponding correction..." is unclear. Rephrase.

AC: We clarified the sentence (see inserted the 1$^{st}$ paragraph of Sect. 2.1.1 here above).

RC1: L174: why using Tmin as a "general" driver? Models often use the daily average temperature.

AC: Indeed, except for the models TPMt, TPMp, PIAGSI, PIA+ and PIA- (Lang et al. 2019, Zani et al. 2020), all compared models use daily average temperature, whereas Jibran (2013) and Lim et al. (2007) describe cold stress to promote leaf senescence. Since cold stress likely relates more to daily minimum temperature, we used the latter in all models. This is also what was done in the currently most recent model comparison (Zani et al. 2020), which can now directly be compared with our comparison. However, we agree that this adaptation of the models CDD, DM1, DM2, SIAM, TDM1, TDM2, PDM1, PDM2, TPDM1, and TPDM2 must be pointed out and discussed, both of which we did in sections 2.2 and 4.5.2.

*Thus, the note to Table 1 in Sect. 2.2 now reads:*
Daily senescence rate responds to the daily drivers minimum temperature (T) and day length (L), following either a monotonically increasing curve (Mon) with cooler temperatures, which may be weakened or amplified with shorter days (Mon− or Mon+), or a sigmoidal curve (Sig). The threshold value is either a constant (Co) or a linear function (Li) of one or two of the following seasonal drivers: site-specific anomaly of (1) spring phenology (a.dSP), (2) growing season index (a.GSI), and (3) daytime net photosynthesis accumulated during the growing season ignoring or considering water limitation constraints (a.Anet and a.Anet–w), as well as the actual (4) leafy season or growing season mean temperature (TLS and TGS), (5) low precipitation index averaged over the leafy season (LPILS), or (6) adapted low precipitation index of the growing season (LPIZa20). Further, the number of free parameters fitted during model calibration and the sources for each model are listed (i.e., De09: Delpierre et al. (2009); Du05: Dufrêne et al. (2005); Ke15: Keenan and Richardson (2015); La19: Lang et al. (2019); Li19: Liu et al. (2019); Za20: Zani et al. (2020)). ==Note that the models CDD, DM1, DM2, SIAM, TDM1, TDM2, PDM1, PDM2, TPDM1, and TPDM2 are originally driven by daily mean rather than daily minimum temperature (cf. Sect. 4.5.2)==. All models are explained in detail in Supplement S2.

*And Sect. 4.5.2 reads:*
==**4.5.2 Daily minimum vs. mean temperature as driver of the senescence rate**==
==Most original publications of the compared models calculated the senescence rate from the mean rather than the minimum temperature, whereas we used the minimum==

temperature for all models. Our choice was based on the stress exhibited by cool temperatures that promotes leaf senescence (Jibran et al., 2013; Lim et al., 2007) and the recent model comparison by Zani et al. (2020), who used daily minimum temperature throughout their study. This choice allowed to compare or study with Zani et al. (2020) and to assess the response curves of the senescence rate. However, inferences on the drivers of leaf senescence would be more profound, if they were based on a comparison that additionally considers models driven by mean temperature, as suggested in some of the original publications (i.e., Delpierre et al., 2009; Dufrêne et al., 2005; Keenan and Richardson, 2015; Liu et al., 2019). Such an extended comparison is certainly essential to gain further insight in the process of leaf senescence but may only focus on the models to remain feasible, rather than also including optimization algorithms and sampling procedures.

RC1: L178-179: 0.25° is a quite coarse spatial resolution, notably when it comes to mountainous areas. Any correction of temperature with altitude (through lapse rate)?

AC: To not further complicate and lengthen the study, we did not correct temperature with elevation. However, we discussed this in the newly added section 4.5.1.

*Sect. 4.5.1 reads:*
4.5.1   Driver data
Modelled weather data can be biased, which affects model outputs based on these data. For example, correcting climate projections for bias increased the accuracies of projected forest ecosystem function and of the simulated timing of leaf phenology (Drepper et al., 2022; Jourdan et al., 2021). Here, we refrained from bias-correcting the meteorological data for the past and future, which likely negatively affected the accuracy of the simulated timing of autumn leaf phenology for the past and future. Thus, we probably received too large RMSE and projected shifts that were both too small and too large. But did the use of uncorrected meteorological data affect our comparison of model vs. calibration effects on model performance and projections? The used meteorological data for the past is likely more accurate for some sites than for others. This is probably also true for the used meteorological data for the future, but the sites with more vs. less accurate data likely differ between climate scenarios. In addition, some scenarios can be systematically warmer than others, for example (cf. above). Therefore, the effect of sites on model performance and the effect of climate scenarios on model projections was probably inflated by the uncorrected meteorological data. In contrast, these data probably affected all models similarly and the sampling procedures randomly, whereas the optimization algorithms remained unaffected. Thus, the use of uncorrected meteorological data most likely had little impact on our results.
Spatial and elevational differences between a particular site and the centre of the corresponding grid cell, from which the meteorological data were extracted, affect the input data. Gridded data may poorly represent the conditions at a particular site due to spatial and elevational differences. For example, precipitation and temperature can change in response to different terrain and the lapse rate, respectively, while the leaf area index and plant-available water capacity can change due to different vegetation and soil conditions. These effects of spatial and elevational differences were not considered in this study and may have led to inaccurate input data (e.g., average MAT for the site Grossarl, 47.2° N / 13.2° E at 900 m a.s.l., in the Austrian Alps was ~0.6° C, which makes beech growth unlikely; Holtmeier and Broll, 2020). The degree of inaccuracy probably differs between sites, which inflated the site effects on model

performance and model projections. In contrast, the effects of models, sampling procedures, and optimization algorithms were probably unaffected by the inaccurate input data (cf. above), so these data most likely had a neglectable effect on our results.

RC1: L183: what is a "climate model chain"? Is one CMC corresponding to one particular climate model run under a particular RCP?

> AC: A CMC is a particular combination of a global and regional climate model (clarified in the revised manuscript).

RC1: Table 1: From Suppl Mat 2, it is unclear how you implemented the relatively complex responses to GSI and Anet described in Ziani et al. 2020. Did you actually code those responses? What cast doubts to me is your use of the term "apparent photosynthesis" in Suppl Mat 2, though it seems from Suppl Mat 1 that you indeed computed photosynthesis. This should be clarified in Suppl Mat 2, e.g. with a mention to Ziani et al. 2020 (their suppl. mat.).

> AC: Yes, we coded these responses according to the Supplement S3 (Sect. S2.2 and S2.3). Following your comment, we clarified that GSI, Anet and Anet–w were calculated by ourselves. (Supplement S2: p. 2 and Supplement S3: pp. 1 and 4) Unfortunately, we are unable to fully understand your doubts as "Suppl Mat" 1 and 2 probably point to Supplements S1 and S2 but these supplements explain the used data (S1) and phenology models (S2). However, we believe you referred to the Supplements S2 and S3 instead, which explain the used models and model drivers, respectively, and answer your comment accordingly. To our understanding of Egle (1960) and Wohlfahrt and Gu (2015), for example, the apparent photosynthesis ($A_{net}$) equals the difference between the real (or gross) photosynthesis ($A_{gd}$) and light (or daytime) respiration. And this is what we calculated in Eq. S32 of Supplement S3, defining light respirations as the daytime fraction of apparent respiration ($R_d$):
> $A_{net} = A_{gd} - R_d \times L / 24$ .
> Our understanding from Zani et al. (2020) is that they also did the following. While they described their models in the main part as dependent on photosynthesis, Eqs. 20 and 22 in their supplement clearly point to the apparent photosynthesis. Therefore, we strongly agree with you that it must be clarified if the apparent or real photosynthesis was used. We now have done so in the changes in section S2.3 of Supplement S3. In addition, we calculated these drivers ourselves, which we also specified in section S2 of Supplement S3.
>
> *Thus, the 1st paragraph of Supplement S3: Sect. 2 now reads:*
> The threshold value is either a constant or depends linearly on one or two of the following seasonal drivers: the timing of spring phenology in the current year, the mean temperature or low precipitation index (*LPI*) of the typical growing season or leafy season, the current growing season index (*GSI*), or the accumulated apparent photosynthetic rate ignoring or considering water limitation constraints ($A_{net}$ or $A_{net-w}$) during the current growing season. These drivers were calculated from the minimum, mean, and maximum air temperature, net short- and longwave radiation, downwelling shortwave radiation, precipitation, and soil moisture as well as plant-available water capacity, atmospheric $CO_2$ concentration, leaf area index and plant functional type (cf. Supplement S1). We calculated driver values according to Eqs. S2–S42 and either applied directly or as site-specific anomalies (in the SIAM, $SIAM_{Za20}$, $PIA_{GSI}$, $PIA^+$, and $PIA^-$ models), depending on the model. Furthermore, all driver values except for the spring phenology depend on the period for which they are calculated. This period

was either from observed spring to average autumn phenology per site and species (average temperature, $LPI_{Za20}$, $GSI$, $A_{net}$, and $A_{net-w}$ for the $TDM_{Za20}$, $PDM_{Za20}$, $TPDM_{Za20}$, $PIA_{GSI}$, $PIA^+$, and $PIA^-$ models) or from observed spring phenology to the first day of the accumulation period ($d_1$; average temperature and LPI for the TDM1, PDM1, TPDM1, TM2, PDM2, TPDM2 models).

*Thus, the 1st paragraph of Supplement S3: Sect. S2.3 now reads:*
The threshold values of the $PIA^+$ and $PIA^-$ models are both driven by apparent photosynthesis, ignoring and considering water limitation constraints, respectively ($A_{net}$ and $A_{net-w}$; [mol m$^{-2}$]; Zani et al., 2020). $A_{net}$ and $A_{net-w}$ were accumulated between observed spring phenology and the site-specific average autumn phenology. They are based on apparent photosynthesis,  This change can be calculated by deducting daytime respiration from gross photosynthesis ($A_{gd}$; [mol m$^{-2}$]), which may also be referred to as light respiration and real photosynthesis, respectively (Egle, 1960; Wohlfahrt and Gu, 2015). $A_{gd}$ is limited by photon availability, Rubisco activity, and sink capacity (Farquhar et al., 1980; Kirschbaum and Farquhar, 1984; Collatz et al., 1991). It can be expressed as min($J_E$, $J_C$, $J_S$), with $J_E$, $J_C$, and $J_S$ being the respective rates depending on light, Rubisco activity, and sink capacity (Collatz et al., 1991, Eq. A.1):

RC1: L248: were the models tested in their ability to simulate trends in observed data (if any over 20-65 years?)

AC: Model accuracy was assessed solely based on the RMSE and we did not perform an additional assessment of the model's ability to simulate past trends. However, this would certainly be an interesting study, which we may conduct in the future.

RC1: L259-260: Does this mean that the parameters used for model evaluation and model projections were possibly different? Why so?

AC: Yes, in site-specific calibration, the parameters were possibly different. The site-specific models were evaluated with a 5-fold cross-validation, which leads to five sets of parameters. Which of these sets should be used for projections? Maybe the set that led to the smallest RMSE or the average parameters calculated from all five sets? On the one hand, different RMSE are likely related to differences in the observations used for validation and thus do not justify the use of one parameter set over another. On the other hand, average parameters are unlikely to lie in any local optimum or even the global optimum. Compared to these alternatives, the selection of parameters derived from the calibration with the whole data appeared most reliable to us.

RC1: L440-441: I'm unsure we are here talking about model external validation. If this is the case, it is not particularly surprising that site-specific calibration can sometimes yield, at the very same site, unrealistic results when the model is used to predict unknown data. Indeed, site-specific parametrisation are more prone to over-fitting (few data points over which to fit the model), as compared to species-specific parametrisation (which includes many sites).

AC: We are not sure, if we understand you correctly. With "external validation" we refer to the fact, that the models were validated with observations that have not yet been used for their calibration. These "external" observations, however, were recorded at the same site as the observations used for calibration. This may lead to unrealistic results due to overfitting, if the number of observations left for calibration is low,

which is exactly what we wanted to study (i.e., effects of the sample size). We agree that such unrealistic results should be expected, why we deleted the word "surprisingly".

*Thus, the 1ˢᵗ paragraph of Sect. 3.1.1 now reads:*
Across the phenology models, optimization algorithms, and sampling procedures, the observed distribution of the external root mean square error (RMSE) of the pooled species differed considerably between the calibration and validation modes. Overall, the smallest median RMSEs were similar between the site- and species-specific calibration modes, ranging from 10.1 to 12.4 and 11.7 to 12.6 or 12.4 to 12.9 days in the respective site- and species-specific calibration validated within sample or within population (Fig. 3, Supplement S6: Table S1). The smallest mean RMSE were considerably larger with the site- than with the species-specific calibration (19.2–52.1 vs. 11.6–23.9 or 12.9–24.4 days; grey dots in Fig. 3). Accordingly, standard deviations were larger with the site- than with the species-specific calibration (28.4–66.5 vs. 3.7–36.6 or 1.2–36.0 days; Fig. 3, Supplement S6: Table S1).

RC1: Fig. 3b: inversion of the x-axis is confusing. It took me several minutes to understand this subplot (as compared to the text description), because i intuitively tried to interpret the x-axis with negative values on the left of the vertical dashed line.

AC: We felt that the plot is easier to understand when more accurate models are visualized with a dot further to the right, but maybe we were wrong. Since we cannot be sure, what our readers will prefer, we changed the plots such that the x-axes are not inversed anymore (L581).

*Thus, Figure 3 now is:*

[Figure]

RC1: L531: how possible is this? Considering that NA-producing and non-converging runs are assigned high RMSE values.

> AC: We agree that this finding for species-specific models is counterintuitive.
> However, on the one hand, the LMMs are not perfect (adjusted $R^2$ range from 0.41 to

0.54) and the ranks of the reference model CDD, whose RMSE is specified on L618 and which has confused you, changed from 14 to 18 and 14 for species-specific models validated within sample and population, respectively. On the other hand, large samples led to most NA-containing calibration runs (Supplement S6: Fig. S1). At the same time, GenSA large samples also led to the lowest RMSE when NA-values were replaced with 170 d. It may well be that many of these NA-producing runs based on large samples led to lower-than-average external RMSEs, especially since the punishing effect of a particular replacement is weaker in large samples. We added a paragraph in section 4.3 (L930-942), in which we discuss this issue.

*Thus, the 4th paragraph of Sect. 4.3 now reads:*
The RMSE increased for some models when non-converging calibration runs or runs that yielded NA values in either the calibration or validation were excluded. For our performance results, we substituted NA values with an error between observed and simulated day of autumn leaf phenology of 170 d (i.e., a larger error than observed in any calibration run). Accordingly, non-converging runs led to an RMSE of 170 d, which were analyzed together with the RMSE of converging runs (cf. Sect. 2.5.2). Now, if the RMSE is analyzed excluding the non-converging runs and the runs that yielded NA values, intuitively one would expect the average RMSE to shrink, but this was not the case in the species-specific models (cf. Supplement 6: Sect. S2.2.2). In other words, punishing with large RMSE and large errors for non-converging runs and NA values led to smaller estimated RMSE and thus better estimated model performance. The relationship between performance and sample size may explain this counterintuitive result. Large samples favored the performance of species-specific models but also more often led to NA values than smaller samples (cf. Supplement 6: Sect. S1). At the same time, larger samples weaken the effect of a particular substitution of an NA value with 170 d on the RMSE. Thus, calibrations with large samples may well have been more accurate despite some NA values and may have resulted in lower RMSE despite NA substitution, which positively affected the overall performance of the models.

RC1: L534-548: a question relative to optimization algorithm is: is the identity of the algorithm involved, or is it more the design of the simulation: in other words we see in SM4, Table S1 that the "normal" vs. "extended" runs of the calibration procedures include less vs. more iterations. I do not see an analysis considering the influence of this number of iterations per algorithm. Apparently, the influence of iteration number is small (Fig 3b: points are close whether considering "norm" or "extd" for one algorithm). If the "extd" simulation number was extended further, would that modify the results?

AC: The influence of the number of iterations per algorithm can be seen in Fig. 3a, for example. More iterations led to better results in most cases except for CMA-ES in site-specific models and for PSO in species-specific models. Since the number of iterations affects the step size with which the global optimum is searched (exploration-exploitation trade-off), we assume that the step size became too small in the afore mentioned exceptions. This led us to the conclusion that careful tuning of the algorithms is very important.
We agree that there was no specific analysis of the influence of this number. Analyzing the effect of only two states, i.e., "few" vs. "many" iterations is certainly not enough. A profound study of the effect of the number of iterations together with a suggestion of how to set the corresponding parameters of the algorithm would be helpful for the modelers.

RC1: Fig 4b: if CMC numbers point to particular climate models, one sees the effect of the climate model can be huge ! Were these climate models unbiased (against local observed climate data)? Climate model bias can influence process-based model simulations strongly, see e.g. Jourdan et al. 2021

> AC: Yes, CMC numbers point to particular models. For time reasons, we refrained from bias correcting these data, but we now discuss this in section 4.5.1 (see further above).

RC1: L621-623: I do not see this on Fig. 4b (i.e. dot coefficients of DM2 and DM2Za20 are not remarkable relative to other models)

> AC: Yes. Here we refer to Supplement S6: Fig. S6; Supplement S6: Table S27.

RC1: L698: rewrite to "phenology"

> AC: Done.

RC1: L711-712: recall which models lead to the best results in Zani et al. 2020

> AC: We already discuss the best models in sect. 4.1 and mention the corresponding original studies. We now referred to Lang et al. (2019) and Zani et al. (2020) as studies with which our findings do not agree.

> *Thus, the 2nd paragraph of Sect. 4.1 now reads:*
> Relatively simple models driven by daily temperature, day length, and partly by seasonal temperature or spring leaf phenology performed best. Patterns in ranked coefficient estimates generally showed that the models DM1 and DM2 developed by Delpierre et al. (2009) and the models DM1$_{Za20}$, DM2$_{Za20}$, TDM$_{Za20}$, and SIAM$_{Za20}$ adapted by Zani et al. (2020) performed best. These models are very similar to the models that Liu et al. (2020) adapted from Delpierre et al. (2009) and Caffarra et al. (2011), which performed best in their model comparison study. Further, the models DM1 and DM2 performed best in Delpierre et al. (2009). However, the models DM1$_{Za20}$, DM2$_{Za20}$, TDM$_{Za20}$, and SIAM$_{Za20}$ did not lead to the best results in Zani et al. (2020). Daily senescence rate in all these models depends on daily temperature and day length, while the threshold is either a constant or linearly derived from the actual average temperature during the growing season or the site-anomaly of spring phenology. Interestingly, the best performing models in site-specific calibration were those adapted by Zani et al. (2020) such that the senescence rate was based on a sigmoid curve, which economized one free model parameter (Table 1 and Supplement S2: Table S1). We hypothesize that fewer parameters generally lead to an advantage with few training observations, which needs to be examined in more detail in further studies. Finally, our study supports previous studies that have also demonstrated the superiority of models based on daily temperature, day length, seasonal temperature, and spring phenology, while questioning the effect of photosynthesis on autumn leaf phenology as suggested by Zani et al. (2020) and indicating that the model by Lang et al. (2019) benefits from considering seasonal drivers.

RC1: L741 "other models": recall briefly their characteristics here

> AC: Done (see the 2nd paragraph of Sect. 4.1 inserted here above).

RC1: L750: rephrase to "local adaptation (Peaucelle et al. 2019)"

    AC: Done.

RC1: L752: "the less such consideration is possible" : unclear, rephrase.

    AC: Done.

*Thus, the 3rd paragraph of Sect. 4.1 now reads:*
Sites had a relatively large influence on projections with species-specific models, and the number of sites per sample had a negative effect on the sample-level model performance of species-specific models. Relevant drivers may be missed by models based on senescence rates driven by temperature and day length and a corresponding threshold (e.g. Fu et al., 2014b). Recent models accounted for this and based their threshold for the senescence rate on spring phenology (SIAM model; Keenan and Richardson, 2015) or on seasonal drivers such as the average growing season temperature or accumulated apparent photosynthetic product (TDM or PIA models; Liu et al., 2019; Zani et al., 2020). However, Gill et al. (2015) and Chen et al. (2018) observed site-specific responses of leaf phenology to climate change, which could be due to site-specific soil properties (Arend et al., 2016), nutrient availability (Fu et al., 2019), and local adaptation (Peaucelle et al., 2019), which are not yet included in the current models. In addition, observations may be biased (Liu et al., 2021) and the perceptions of observers at different sites are usually not aligned. Models can consider relevant but excluded drivers and observer bias through differently calibrated parameters, such as a sample-specific threshold constants, for example. this is not possible to the same extent for species-specific models as for site-specific models. However, the positive effect of such sample-specific parameters decreases as the number of sites in the sample increases, shifting the modeled values closer to the mean observation of the calibration sample. Consequently, we expect (1) a larger effect of sites on projections based on species-specific rather than site-specific models and (2) an increasing RMSE with more sites per sample. Here we observed both, i.e. the relative influence of sites on projections increased from 24% to 46% or 41% if based on site-specific models or species-specific models projected within sample or within the entire population, respectively. Moreover, the RMSE of species-specific models validated within sample increased with more sites as expressed by the site ratio s:S. This demonstrates that some relevant drivers of autumn leaf phenology are not yet considered in the evaluated process-oriented models and/or that bias in observed phenology data may be amplified when the same observer is at a particular site for multiple years.

RC1: L758-759: and/or that the observed data are more prone to observation bias, that can magnify if same observer is operating across years at a given site (in practice, observers inter-calib are rare). See Liu et al. 2021

    AC: We added corresponding thoughts to Sect. 4.1 (see the 3rd paragraph of Sect. 4.1 inserted here above).

RC1: L788-789: interesting result. Where does this "17 sites" come from? I do not remember seeing that earlier in the manuscript.

AC: 17 is the number of sites in stratified samples based on the average timing of autumn phenology (L270). The result is mentioned in L454–L465, visualized in Figure 3d, and quantified in Supplement 6: Table S2.

RC1: L819: Cochran (1946) is missing from the reference list

AC: We now included this reference.

RC1: L852: remove "but see"

AC: Done.

RC1: L889-891: we touch here the question of local adaptation again

AC: Yes. We now mentioned this explicitly in L991.

*Thus, the 3rd paragraph of Sect. 4.4 now reads:*

Sites generally exhibited the second largest influence on projected shifts and had more influence when the latter were projected with species- than with site-specific models, which could be due to correct modelling of site differences or to poorer calibration for sites with phenology that deviates from the sample mean. Studies of past changes in autumn phenology of trees found ambiguous trends between different sites (Piao et al., 2019; Meier et al., 2021). Different trends may be the result of opposing weather effects, e.g. moderate versus severe heat and drought spells (Xie et al., 2015), or of different correlations between spring and autumn phenology, dependent on nutrient availability and elevation (Fu et al., 2019; Charlet de Sauvage et al., 2022), possibly related to local adaptation (Alberto et al., 2011; Quan and Wang, 2018). Thus, it may be that such opposing weather effects or different correlations led to the strong influence of sites on projected autumn phenology. However, this strong influence could also be due to the above-described tendency of models to the mean and subsequent poorer calibration for sites at the extremes of the climatic and phenological spectrum within samples. This hypothesis is further supported by the larger relative influence of sites on projections with species- than with site-specific models. In addition, species-specific models performed worse than site-specific models (based on converged runs without NA values), as was also reported by Basler (2016) with respect to models of spring phenology, while Liu et al. (2020) found that species-specific models poorly reflected the spatial variability of autumn phenology. Thus, it seems improbable that the species-specific models with generally poorer performance predicted site differences more accurately than the site-specific models with generally better performance. Therefore, we suspect that the large influence of sites on projections with species-specific models is primarily due to insufficiently modelled processes of leaf phenology and the consequent tendency of phenology models to the mean.

RC1: L897: "Different models altered the reference shifts by -12 to +2 days", recall the average.

AC: Done.

RC1: L970: "... and found our data to strongly encourage further research" is unclear.

AC: We tried to refer to the exploratory nature of our study. Since we mentioned this later, we simply deleted it here.

**References of authors' answer:**

Egle, K. 1960. Apparente und reelle Photosynthese, Gaswechselgleichgewicht, Lichtatmung. Pages 182-210 *in* A. Pirson, editor. Die CO2-Assimilation / The Assimilation of Carbon Dioxide: In 2 Teilen / 2 Parts. Springer Berlin Heidelberg, Berlin, Heidelberg.

Jibran, R., D. A. Hunter, and P. P. Dijkwel. 2013. Hormonal regulation of leaf senescence through integration of developmental and stress signals. Plant Molecular Biology **82**:547-561.

Lang, W., X. Chen, S. Qian, G. Liu, and S. Piao. 2019. A new process-based model for predicting autumn phenology: How is leaf senescence controlled by photoperiod and temperature coupling? Agricultural and Forest Meteorology **268**:124-135.

Lim, P. O., H. J. Kim, and H. Gil Nam. 2007. Leaf senescence. Annual Review of Plant Biology **58**:115-136.

Wohlfahrt, G., and L. Gu. 2015. The many meanings of gross photosynthesis and their implication for photosynthesis research from leaf to globe. Plant, cell & environment **38**:2500-2507.

Zani, D., T. W. Crowther, L. Mo, S. S. Renner, and C. M. Zohner. 2020. Increased growing-season productivity drives earlier autumn leaf senescence in temperate trees. Science **370**:1066-1071.